# Peak and Cumulative Response of Reinforced Concrete Frames with Steel Damper Columns under Seismic Sequences

Kenji Fujii

Department of Architecture, Faculty of Creative Engineering, Chiba Institute of Technology, Chiba 275-0016, Japan; kenji.fujii@p.chibakoudai.jp

**Abstract:** The steel damper column is an energy-dissipating member that is suitable for reinforced concrete (RC) buildings and those used for multistory housing in particular. However, the effectiveness of steel damper columns may be affected by the behavior of surrounding members, and this effect can be severe in the case of seismic sequences. This article investigates the nonlinear response of building models with an RC moment-resisting frame (MRF) with and without steel damper columns under seismic sequences. The applicability of the concept of the momentary energy input to the prediction of the peak response of RC MRFs with damper columns under seismic sequences is also investigated. The main findings of the study are summarized as follows. (1) The peak response of RC MRFs with damper columns subjected to sequential accelerations is similar to the peak response obtained considering only the mainshock, whereas the cumulative strain energy of RC MRFs accumulates more for sequential accelerations. (2) The steel damper column is effective in reducing the peak and cumulative responses of RC MRFs in the case of sequential seismic input. (3) The relation of the hysteretic dissipated energy during a half cycle of the structural response and the peak displacement of the first modal response can be properly evaluated using the simple model proposed in this study.

**Keywords:** reinforced concrete moment-resisting frame; steel damper column; seismic sequence; peak response; cumulative response; cyclic degradation; passive control structure; momentary energy input

## 1. Introduction

### 1.1. Background

In an earthquake-prone country such as Japan, controlling seismic damage to a structure is an important issue in the seismic design of building structures. A popular and classical strategy for improving the damage control ability of the moment-resisting frame (MRF) is the so-called weak-beam strong-column concept. This strategy is widely accepted and is recommended for the seismic design of MRFs. When this strategy is adopted, most of the seismic energy is absorbed by plastic hinges set at each beam end. However, because the beams also carry gravitational loads, those MRF buildings may not continue to be usable after a huge earthquake due to the severe damage to their beams. In addition, as was the case in the 2016 Kumamoto earthquake, there may be a sequence of large foreshocks and the mainshock or a sequence of the mainshock and large aftershocks. In such cases, seismic energy accumulates at the plastic hinges and causes damage. Therefore, an MRF designed solely according to the weak-beam strong-column concept may be insufficient in the case of such seismic sequences. A dual system with sacrificial members that absorb the seismic energy prior to the beams and columns (e.g., in a damage-tolerant structure) [1] is one solution for creating structures with superior seismic performance.

The steel damper column [2] is an energy-dissipating sacrificial member. Figure 1 compares the design collapse mechanism of the traditional MRF and MRF with steel damper columns. In the traditional MRF, shown in Figure 1a, most of the seismic energy is absorbed at the plastic hinges of the beam ends and the bottom end of the first story columns.

Meanwhile, for the MRF with steel damper columns shown in Figure 1b, the damper panel within a damper column absorbs seismic energy prior to the plastic hinges in the beams and columns. The energy absorbed by the plastic hinges can thus be reduced using steel damper columns.

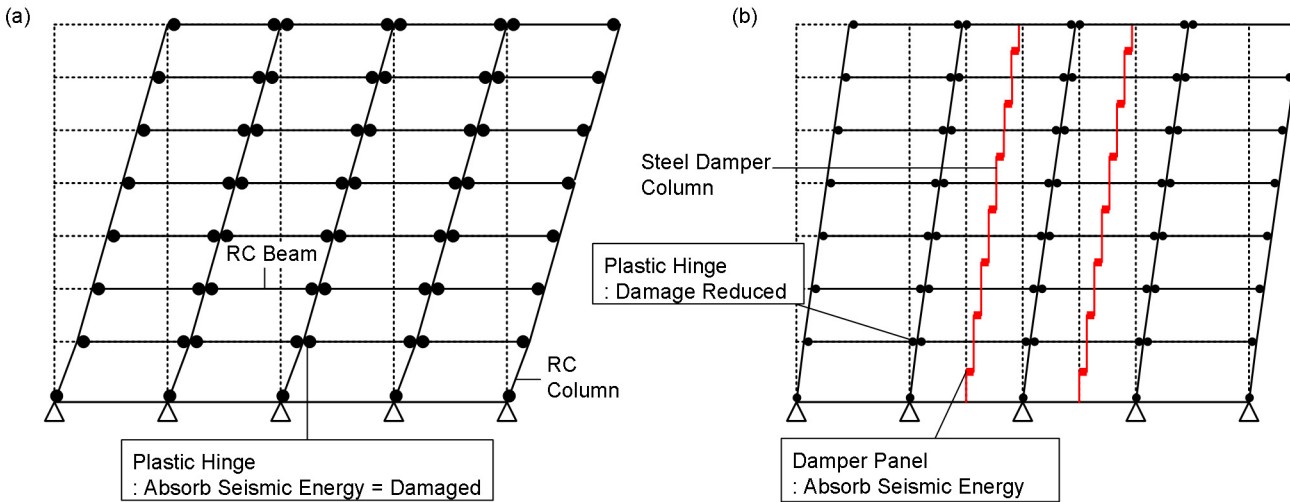

**Figure 1.** Design collapse mechanism of MRFs: (**a**) traditional MRF designed according to the strong-column weak-beam concept; (**b**) MRF with steel damper columns.

Figure 2 shows the design concept of a reinforced concrete (RC) MRF with steel damper columns. Unlike the use of buckling-restrained braces, the use of steel damper columns provides usable space for corridors, as shown in Figure 2a. The steel damper column is thus suitable for high-rise RC housing in that it minimizes obstacles in architectural planning. Figure 2b presents an example of the RC beam–steel damper column joint. The steel beam embedded in the RC beam transfers the bending moment from the RC beam to the damper column.

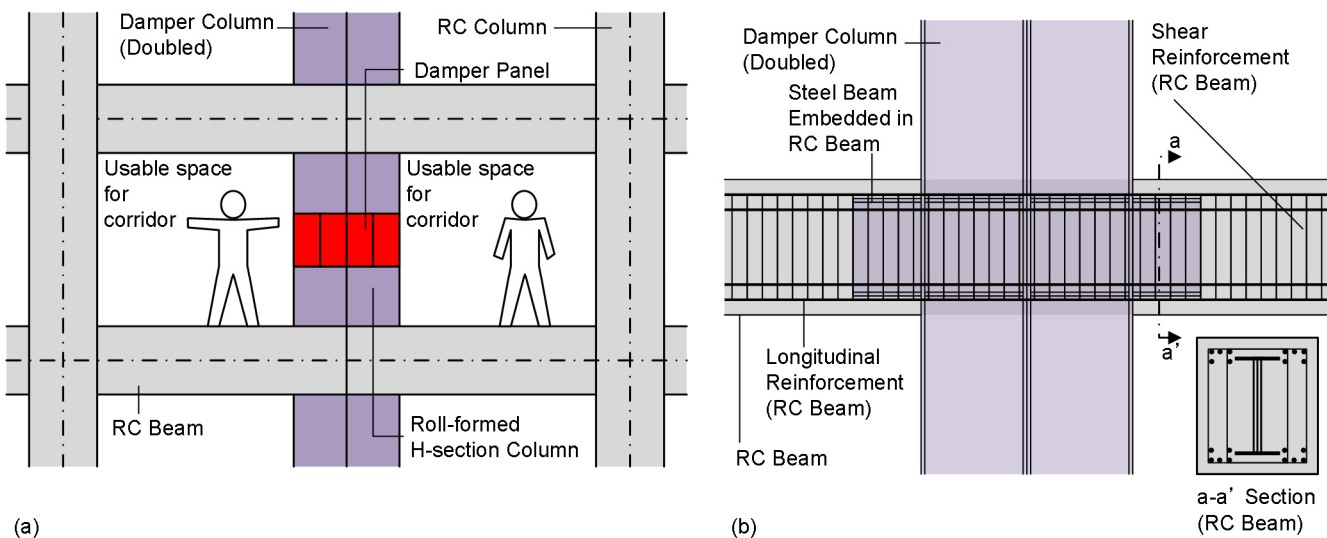

**Figure 2.** Design concept of the RC MRF with steel damper columns: (**a**) conceptual structural elevation; (**b**) details of the RC beam–steel damper column joint.

The author previously investigated the development of the rational seismic design procedure of an RC MRF with steel damper columns [3–5]. Specifically, a displacement-controlled seismic design method was proposed for an RC MRF with steel damper columns [3,4]. With this method, the strength demand of the whole structure is determined through

equivalent linearization. Although the method is simple, it does not consider the accumulated strain energy of members (both RC members and steel damper columns) or the effect of seismic sequences. Another important issue with an RC MRF with steel damper columns is that the effect of the steel damper columns depends on the behavior of the surrounding RC beams. The author previously found [5] that a proper strength balance of the steel damper columns and surrounding beams is important in maximizing the dissipation of energy into the damper columns; i.e., the beam-end section connected to the damper columns requires sufficient strength to avoid premature yielding prior to any energy dissipation.

The previous results reveal the importance of studying the nonlinear seismic behavior of an RC MRF with steel damper columns designed using the previously proposed method [3] under seismic sequences and the effectiveness of steel damper columns for the reduction of structural damage.

### 1.2. Brief Review of Related Studies
### 1.2.1. Studies on the Responses of Structures under Seismic Sequences

Many studies have investigated the response of structures under seismic sequences (e.g., [6–50]). To the best of the knowledge of the author, the first study on the nonlinear response of structures to seismic sequences was conducted by Mahin [6]. Later, Amadio et al. investigated the nonlinear response of the idealized single-degree-of-freedom (SDOF) model and two-dimensional frame model under repeated earthquake ground motions [7]. In their study, identical ground motions were applied several times as the seismic sequence. Hatzigeorgiou studied the nonlinear response of an SDOF model [8–10] and RC frame models [11–13]. In those studies, the artificial seismic sequences were created by selecting ground motions at random, neglecting the difference in the frequency characteristics of the mainshock and aftershocks (or foreshocks). Ruiz-García et al. pointed out the problem with the assumptions made in creating artificial seismic sequences [14–22]. Specifically, they showed that the predominant period of aftershocks is shorter than that of the mainshock because the magnitudes of the aftershocks are smaller than the magnitude of the mainshock (e.g., [14,15,17]). They thus concluded that using artificial seismic sequences with repeating identical accelerations may lead to the overestimation of the effects of seismic sequences [15]. In addition, they pointed out that the ratio of the predominant periods of aftershocks and the mainshock is an important parameter in discussing the effect of a seismic sequence [14,16,18,20]. They also conducted three-dimensional frame analyses considering seismic sequences [19,22]. Similar findings were obtained by Goda et al. [23–26], who proposed a method of generating artificial ground motion sequences by considering the difference in magnitude between the mainshock and aftershocks [23]. They also pointed out the importance of the record selection of aftershocks in incremental dynamic analysis [25]. Tesfamariam et al. studied the seismic vulnerability of RC frames with unreinforced masonry infill due to mainshock–aftershock sequences [27]. Tesfamariam and Goda proposed the seismic performance evaluation framework considering maximum and residual inter-story drift ratio of on-code conforming RC buildings [28] and also energy-based seismic evaluation method of tall RC buildings [29]. In [28], the authors concluded that although the influence of the mainshock–aftershock sequences to the maximum inter-story drift is limited, the seismic performance of non-code-conforming RC buildings are influenced by the mainshock–aftershock sequences because the residual inter-story drift of RC buildings increases due to the seismic sequences. Similarly, they concluded that the influence of major aftershocks on the damage potential is significant because the energy-based damage index of tall RC buildings increases due to the seismic sequences. In addition to these studies, Zhai et al. [30–32], Di Sarno et al. [33–36], Abdelnaby et al. [37–41], Kagermanov and Gee [42,43], Yang et al. [44,45], Yaghmaei-Sabegh et al. [46], Qiao et al. [47], Orlacchio et al. [48], Hoveidae and Radpour [49], and Pirooz et al. [50] investigated the nonlinear response of building structures under seismic sequences. Although most of those studies were analytical in nature, one experimental study used a shaking table [47].

From the author's viewpoint, few studies have investigated the application of the pushover-based procedure using a nonlinear equivalent SDOF model for the seismic evaluation of a building considering seismic sequences; the existing studies are those of Guerrero et al. [21], Kagermanov and Gee [42,43], and Orlacchio et al. [48]. Such studies are essential in the author's opinion. One reason is that the analysis of the nonlinear response of structures using the equivalent SDOF model can lead to a better understanding of the nonlinear behavior of structures. As an example, relations between the seismic intensity parameter and response quantities (e.g., the peak response and cumulative energy) can be clearly discussed using an equivalent SDOF model. In addition, from a practical point of view, such pushover-based procedures provide structural designers and analysts with basic information on the nonlinear behavior of the analyzed building.

Another issue to be addressed is that there have been few studies on structures with dampers, with the existing studies being those of Guerrero et al. [21], Yang et al. [45], and Hoveidae [49]. As described above, the main motivation for installing such dampers is the energy absorption of the sacrificial members prior to beams and columns. It is therefore essential to discuss the effect of dampers in terms of the cumulative strain energy in the event of seismic sequences. However, few discussions have been presented in the studies cited above.

### 1.2.2. Studies on the Seismic Energy Input

The quantification of structural damage to members, such as RC beams, columns, and dampers, is an important issue in rational seismic design and evaluation. Several indices of structural damage have been proposed, e.g., the Park–Ang index, which is defined as the combination of the peak deformation and cumulative energy [51]. Because the cumulative energy is directly evaluated from the seismic energy input, it is rational to consider the seismic intensity according to energy-related parameters. The total input energy [52,53] is a seismic intensity parameter related to the cumulative strain energy. Several studies have investigated total input energy spectra (e.g., [54–58]).

Inoue and his team proposed the maximum momentary input energy [59–61] as an energy-related seismic intensity parameter related to the nonlinear peak displacement. They predicted the peak displacement by equating the maximum momentary input energy and hysteretic dissipated energy in a half cycle of the structural response. The definition of the momentary input energy is described in Appendix A.

Following their work, the present author formulated the time-varying function of the energy input using a Fourier series [62]. This formulation shows that two seismic intensity parameters, namely the maximum momentary input energy and total input energy, can be evaluated based on the properties of the system and the complex Fourier coefficient of the ground motion. The concept of the maximum momentary input energy has been extended for bidirectional excitation [63] and implemented in the prediction of the peak and cumulative responses of a one-mass, two-degree-of-freedom model representing a ductile RC structure [64] and an irregular base-isolated building subjected to bidirectional horizontal ground motions [65]. The application of the momentary energy input to the RC MRF subjected to seismic sequences is thus promising.

### *1.3. Objectives*

Against the above background, the following questions are addressed in this article.

- What are the differences in the peak and cumulative responses of RC MRFs with and without steel damper columns between a single acceleration and sequential accelerations?
- Is the steel damper column effective in reducing the peak and cumulative responses of an RC MRF in the event of a seismic sequence?
- In the prediction of the peak response of the RC MRF with steel damper columns based on the momentary energy input, the relation between the hysteretic dissipated energy during a half cycle of the structural response and the peak displacement must

be properly modeled. How can this relationship be modeled from the results of pushover analysis?

The present article investigates the nonlinear response of 10-story RC MRF building models with steel damper columns designed according to the previously proposed method [3] in a case study of such an RC MRF with steel damper columns subjected to seismic sequences. Ground-motion records obtained from three stations managed by the National Research Institute for Earth Science and Disaster Resilience (NIED) during the foreshock and mainshock of the 2016 Kumamoto earthquake [66] are used. The effects of the seismic sequences on the peak and cumulative responses of the RC MRFs are investigated using the results of nonlinear time-history analysis. The first modal response of RC MRFs is then calculated from the results of the nonlinear time-history analysis, and the applicability of the momentary input energy to the prediction of the peak response under a seismic sequence is discussed.

The remainder of the article is organized as follows. Section 2 presents four RC MRF building models with and without damper columns as well as the ground motion data used in the nonlinear time-history analysis. The results of nonlinear time-history analysis are presented and discussed in Section 3. This section further analyzes the peak and cumulative responses of the RC MRF and the effectiveness of the steel damper column. The first modal response is evaluated using the results of nonlinear time-history analysis and discussed in Section 4. Conclusions and future directions of study are discussed in Section 5.

## 2. Building and Ground Motion Data

### 2.1. Building Data

Figure 3 shows the simplified structural plan and elevation of the MRF building models with steel damper columns considered in this study. Two 10-story building models, namely RCDC1 and RCDC2, are designed using the simplified design method [3,4]. Specifically, RCDC1 is the building model presented in the previous study [3], and RCDC2 is the building model obtained by shortening the span of the RC beam, while the change in the total strength of the overall building model (comprising both the RC frame and damper column) is minimized [4]. The model RCDC1 represents an MRF that has been intentionally designed to be flexible, whereas the model RCDC2 represents an ordinary MRF. The unit mass per floor is assumed to be 1.2 $t/m^2$. The story height is assumed to be 4.5 m for the first story and 3.2 m for upper stories. Details of the members are given in the Appendix B.

To investigate the effect of the steel damper column in reducing the seismic response, two other models are considered by removing all steel damper columns from RCDC1 and RCDC2. Here, the models with dampers removed from RCDC1 and RCDC2 are, respectively, referred to as BareRC1 and BareRC2. Four MRF building models are thus considered in this study.

Figure 4 shows the modeling of the MRF with steel damper columns. In this study, all RC MRFs are designed according to the weak-beam, strong-column concept. A potential hinge is set at all RC beam ends (except the beam end connected to the steel damper columns) and the bottom end of the first-story columns as shown in Figure 4a.

The building is modeled as having a planar frame, as shown in Figure 4b. All RC members are modeled as an elastic beam with a nonlinear flexural spring at both ends. The steel damper columns are modeled as an elastic column with a nonlinear damper panel at the middle of the steel damper column. The beam-column joints are assumed to have rigid behavior. For RC beam–RC column joint, the proper reinforcement is assumed to be provided to prevent premature failure. In addition, the proper dimensions (section sizes and lengths) of steel beam embedded in RC beams and proper reinforcement is assumed to be provided in RC beam–damper column joint, to prevent premature failure until the damper panels reach their ultimate stage.

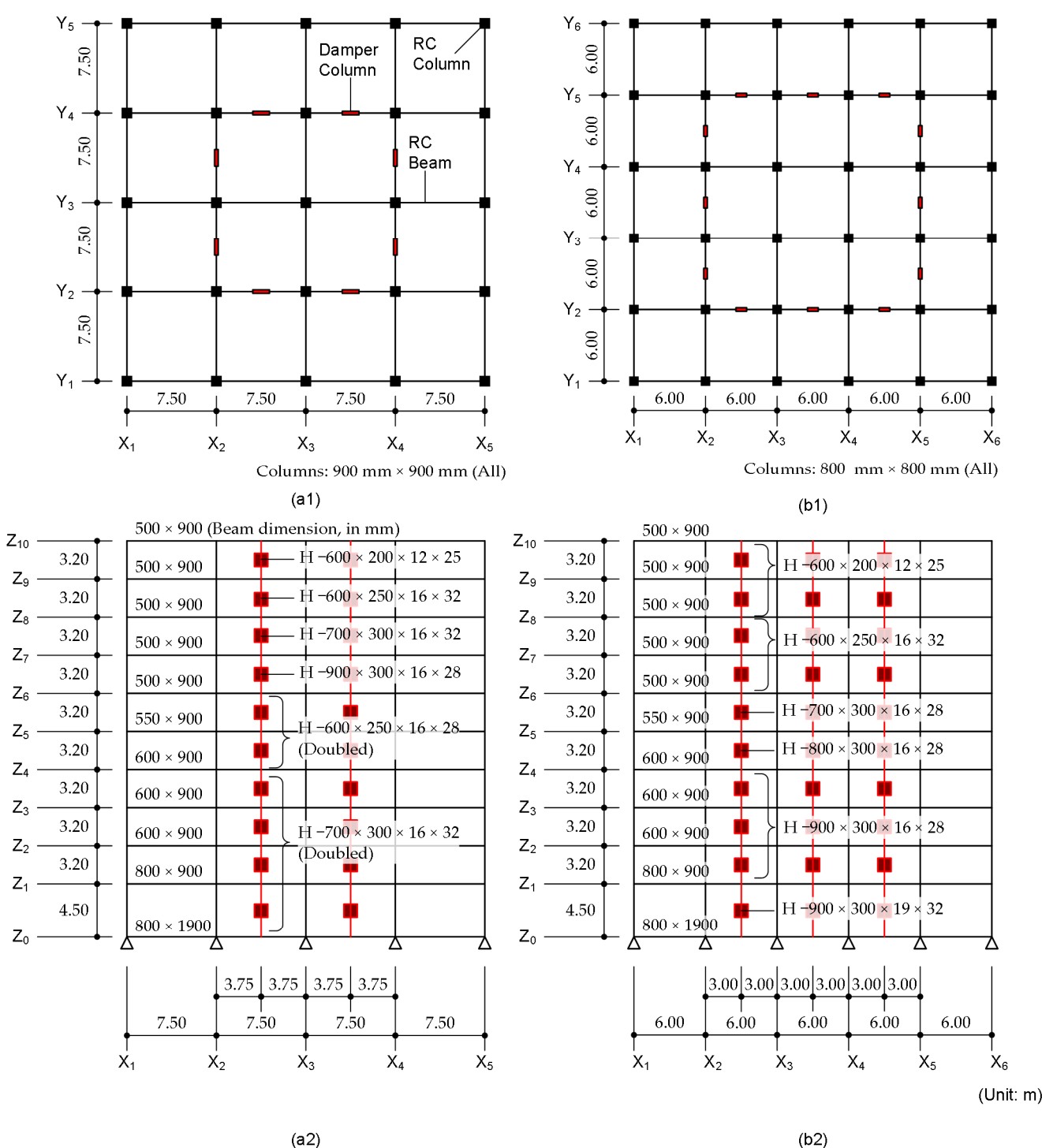

**Figure 3.** RC MRF building models with damper columns: (**a1**) structural plan of RCDC1; (**a2**) structural elevation of RCDC1 (frame Y$_2$); (**b1**) structural plan of RCDC2; (**b2**) structural elevation of RCDC2 (frame Y$_2$).

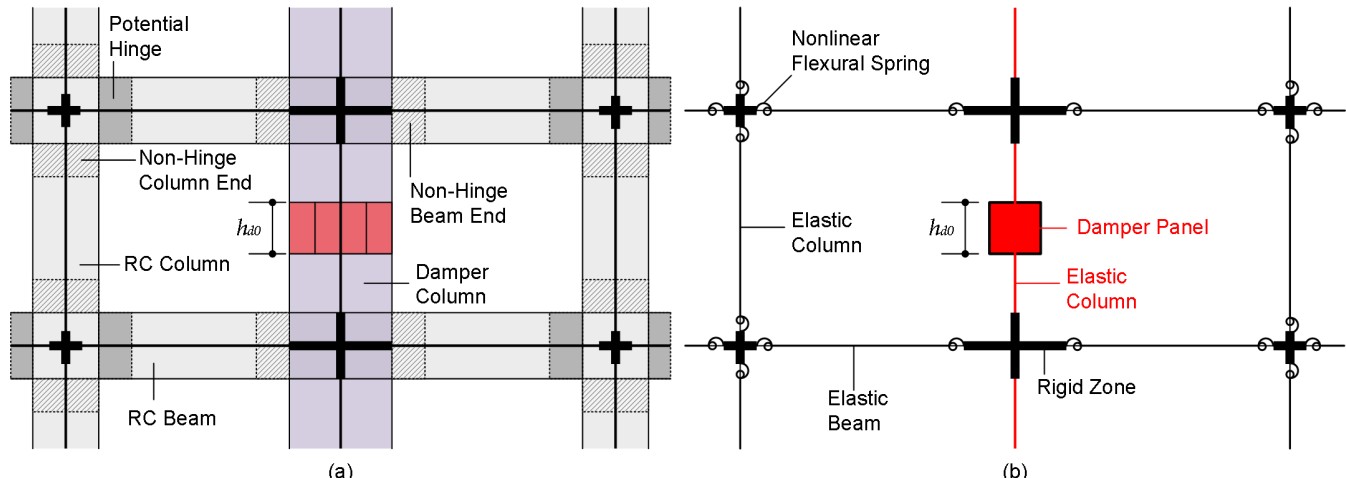

**Figure 4.** Modeling of the MRF with a damper column: (**a**) potential hinge and non-hinge at the beam end and (**b**) structural model.

Figure 5 shows the envelope of the force–deformation relationship of each member. The modeling of each member is the same as that in the previous study [3] and is summarized as follows. The envelopes are assumed to be symmetric under positive and negative loading. At the potential hinge of RC members, the crack and yielding of the section are considered as shown in Figure 5a. The yield moment $M_y$ (at point Y) is calculated according to the AIJ standard [67], whereas the crack moment $M_c$ (at point C) is assumed to be one-third of $M_y$. The secant stiffness degradation ratio at point Y ($\alpha_y$) is calculated using the equation proposed by Sugano and Koreishi [68]. Meanwhile, at the non-hinge ends of RC members, only the cracking of the section is considered, as shown in Figure 5b, except for the beam at the foundation level. The cracking moment ($M_c$) of the non-hinge beam end is assumed to be the same as that at the opposite end, and the tangent stiffness degradation ratio after cracking ($\alpha_1$) is assumed to be the same as the secant stiffness degradation ratio at yielding ($\alpha_y$) calculated following Sugano and Koreishi [68]. Meanwhile, at the non-hinge column end, the cracking moment ($M_c$) is calculated considering the axial force attributed to the vertical load, and the tangent stiffness degradation ratio after cracking ($\alpha_1$) is assumed to be 0.2. The flexural behavior of a beam at the foundation level is assumed to be linearly elastic. The shear behavior is assumed to be linearly elastic of all RC members. In nonlinear static analysis, the bilinear envelope shown in Figure 5c is assumed for the damper panel. Here, $Q_{yDL}$ and $Q_{yDU}$ respectively denote the initial and upper bound yield strengths of the damper panel. The axial behavior is assumed to be linearly elastic for all vertical members.

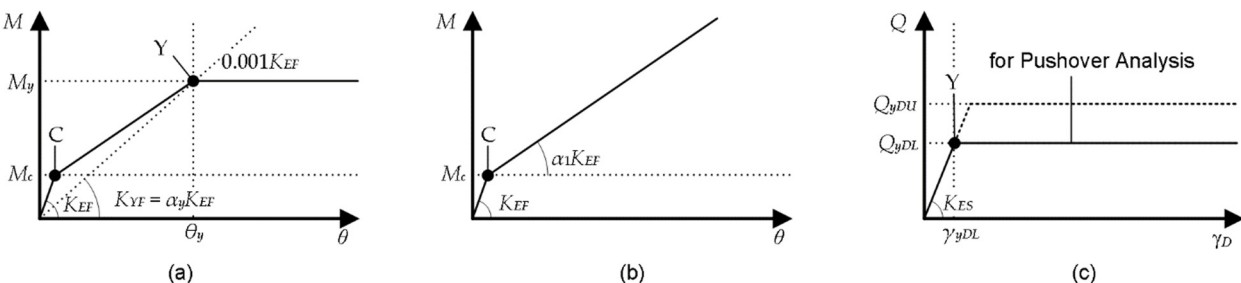

**Figure 5.** Envelope of the force–deformation relationship of each member: (**a**) flexural behavior of the potential hinge of RC members; (**b**) flexural behavior of non-hinge ends of RC members; (**c**) shear behavior of the damper panel in the steel damper column.

Figure 6 shows the hysteresis rule of the nonlinear spring. In this study, the Muto model [69] with two modifications is used for the flexural spring in RC members. The first modification is the unloading of the stiffness after yielding to represent the degradation

of RC members after yielding, as shown in Figure 6a. The second modification is the consideration of stiffness degradation after yielding due to cyclic loading. In this study, the model proposed by Umemura et al. [70] is implemented with the Muto model, as shown in Figure 6a. A parameter $\chi$ is introduced to represent the effect of cyclic degradation. The value of $\chi$ can be taken as zero or positive. In the case that $\chi$ is zero, there is no cyclic degradation, as shown in Figure 6a, which corresponds to the model used in the previous study (e.g., [3,5,64]). In the case that $\chi$ is positive, there is stiffness degradation due to the reloading target point shifting from point $P_p$ to $P_n$, as shown in Figure 6a. Umemura et al. [70] showed that $\chi$ depends on (i) the compressive strength of the concrete, (ii) the shear reinforcement ratio, (iii) the compressive stress of the section normalized by the compressive strength of the concrete, and (iv) the shear-span-to-depth ratio of the member. In this study, the value of $\chi$ is set as given in Table 1 following Umemura et al. [70]. Note that a larger value of $\chi$ is set for the short-span beam; i.e., the stiffness degradation of the short-span beam is more severe than that of the long-span beam. Additionally, it is noted that the cyclic stiffness degradation is more severe in RCDC2 than in RCDC1.

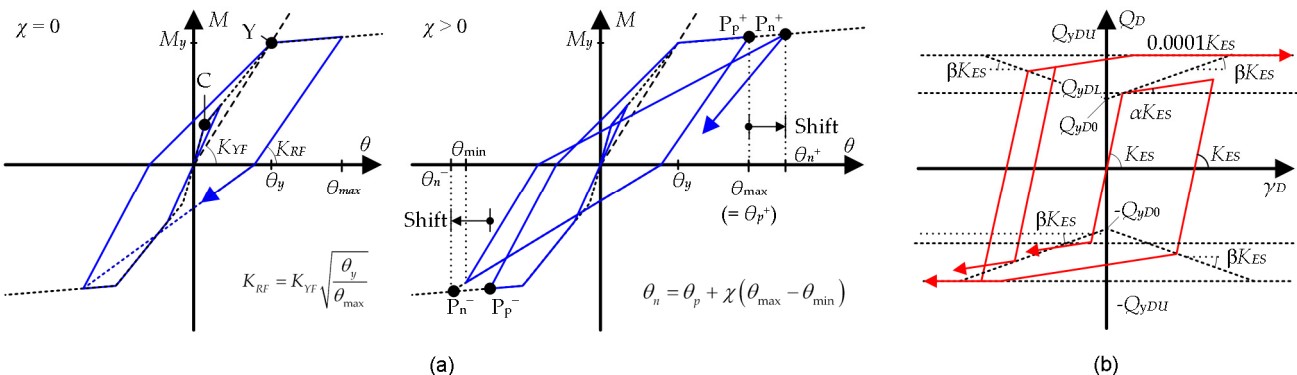

(a)                                                     (b)

**Figure 6.** Hysteresis rule of each member: (**a**) flexural behavior of RC members and (**b**) shear behavior of the damper panel in the steel damper column.

**Table 1.** Values of parameter $\chi$ representing the cyclic degradation of RC members.

|  | BareRC1, RCDC1 | BareRC2, RCDC2 |
|---|---|---|
| Short-span Beam (beams connected to damper column) | $\chi = 0.074$ (only RCDC1) | $\chi = 0.081$ (only RCDC2) |
| Long-span Beam (other beams) | $\chi = 0.034$ | $\chi = 0.049$ |
| Column (bottom-end of the first story) | $\chi = 0.060$ | $\chi = 0.046$ |

It should be mentioned that the pinching behavior observed in RC members is not considered in this study. There are several hysteresis models that implement the pinching behavior, e.g., the model proposed by Baker and Noori [71]. Such pinching behavior may affect the response of damper columns. However, this issue would be the next phase of this study.

The model proposed by Ono and Kaneko [72] shown in Figure 6b is used to model the hysteresis behavior of the damper panel. In this model, the strain-hardening effect of the low-yield-strength steel is controlled by the two parameters $\alpha$ and $\beta$. Here, the two parameters are set as $\alpha = 0.022$ and $\beta = 0.013$. Note that the model has been verified by comparing the test results of shear damper panel and braces using low-yield-strength steel in the literature [72], and this model is implemented in several commercial computer programs used for the actual structural design in Japan. It should be mentioned that Vaiana et al. proposed a more generic hysteretic model for rate-independent mechanical systems [73]. They have applied their models for the modeling of rate-independent passive

energy-dissipation devices [74]: the calibration of their model has been made by comparisons of the experimental tests of steel shear link tested by Nuzzo et al. [75]. Therefore, such a model would be more suitable for the hysteresis model of damper panel than the model applied herein. However, this issue is out of scope of this study.

The damping matrix is assumed to be proportional to the instantaneous stiffness matrix without a damper column. The damping ratio of the first elastic mode of the model without a damper column is assumed to be 0.03. Second-order effects, including the P-Δ effect, are neglected in this study. The soil–structure interaction (SSI) effect is neglected for the simplicity of the analysis.

Table 2 gives the natural periods of the first three modes in the initial stage for each model. As shown here, the natural period of RCDC1 is longer than that of RCDC2.

**Table 2.** Natural periods of the first three modes in the initial stage.

|  | **BareRC1** | **BareRC2** | **RCDC1** | **RCDC2** |
|---|---|---|---|---|
| $T_{1e}$ (s) | 0.8547 | 0.7106 | 0.7044 | 0.6152 |
| $T_{2e}$ (s) | 0.2834 | 0.2427 | 0.2344 | 0.2094 |
| $T_{3e}$ (s) | 0.1577 | 0.1369 | 0.1330 | 0.1195 |

In examining the nonlinear behavior of the four models, pushover analysis (i.e., displacement-based mode-adaptive pushover analysis [76]) is carried out to obtain the relationship between the equivalent acceleration $A_1{}^*$ and equivalent displacement $D_1{}^*$. Let $_n\mathbf{f_R}$ and $_n\mathbf{d}$ be the restoring force vector and horizontal displacement vector, respectively, of the building model at each loading step $n$ obtained in the pushover analysis. The equivalent displacement and acceleration at step $n$ (namely $_nD_1{}^*$ and $_nA_1{}^*$) are determined from Equations (1) and (2), respectively, assuming that the vector $_n\mathbf{d}$ is proportional to the first mode vector $(_n\Gamma_{1\mathbf{n}}\boldsymbol{\varphi}_1)$ at each loading step:

$$_nD_1{}^* = \frac{_n\Gamma_{1\mathbf{n}}\boldsymbol{\varphi}_1{}^\mathbf{T}\mathbf{M_n d}}{_nM_1{}^*} = \frac{_n\mathbf{d}^\mathbf{T}\mathbf{M_n d}}{_n\mathbf{d}^\mathbf{T}\mathbf{M1}}, \tag{1}$$

$$_nA_1{}^* = \frac{_n\Gamma_{1\mathbf{n}}\boldsymbol{\varphi}_1{}^\mathbf{T}{}_n\mathbf{f_R}}{_nM_1{}^*} = \frac{_n\mathbf{d}^\mathbf{T}{}_n\mathbf{f_R}}{_n\mathbf{d}^\mathbf{T}\mathbf{M1}}, \tag{2}$$

$$_nM_1{}^* = {}_n\Gamma_{1\mathbf{n}}\boldsymbol{\varphi}_1{}^\mathbf{T}\mathbf{M1} = \frac{\left(_n\mathbf{d}^\mathbf{T}\mathbf{M1}\right)^2}{_n\mathbf{d}^\mathbf{T}\mathbf{M_n d}} \tag{3}$$

$$_n\mathbf{d} = \left\{ \begin{array}{ccc} _ny_1 & \cdots & _ny_N \end{array} \right\}^T, \tag{4}$$

$$_n\mathbf{f_R} = \left\{ \begin{array}{ccc} _nf_{R1} & \cdots & _nf_{RN} \end{array} \right\}^T, \tag{5}$$

$$\mathbf{M} = \begin{bmatrix} m_1 & & 0 \\ & \ddots & \\ 0 & & m_N \end{bmatrix}, \tag{6}$$

$$\mathbf{1} = \left\{ \begin{array}{ccc} 1 & \cdots & 1 \end{array} \right\}^T. \tag{7}$$

In Equations (1)–(6), $\mathbf{M}$ is the mass matrix, $_nM_1{}^*$ is the effective first modal mass at loading step $n$, and $m_j$ is the floor mass of the $j$th floor. Figure 7 shows the obtained $_nA_1{}^*-{}_nD_1{}^*$ relationship for each model. In the figure, the point labeled "design target" is the point assumed as the displacement limit $D_1{}^*_{limit}$ (=0.2833 m), which is assumed to be 1/82.5 of the assumed equivalent height $H_1{}^*$ (=23.37 m). As shown in Figure 7a,b, the displacement upon the first yielding of the RC member is larger for BareRC1 than for BareRC2. A similar observation is made in the comparison of Figure 7c,d, whereas the displacement upon the first yielding of the damper column is smaller for RCDC1 than for RCDC2.

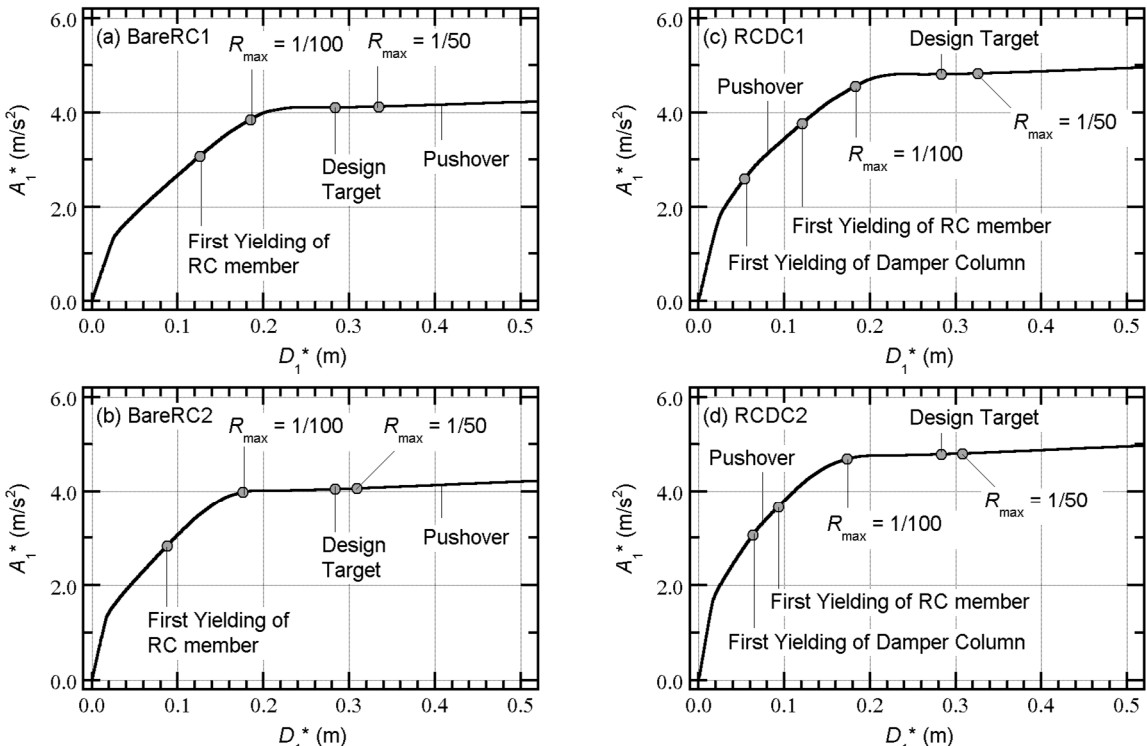

**Figure 7.** Relationship between the equivalent acceleration $_nA_1{}^*$ and equivalent displacement $_nD_1{}^*$ of each model: (**a**) BareRC1; (**b**) BareRC2; (**c**) RCDC1; (**d**) RCDC2.

### 2.2. Ground Motion Data

The present study uses records of accelerations of the foreshock (event time: 14 April 2016, 21:26 (JST), JMA magnitude 6.5) and mainshock (event time: 16 April 2016, 01:25 (JST), JMA magnitude 7.3) obtained at three stations managed by NIED, namely K-NET Kumamoto (KMM), K-NET Uto (UTO), and KIK-NET Mashiki (MAS). Table 3 lists the ground motions, and Figure 8 presents the primary and shear wave profiles for the different stations. The soil properties for each station are available from the K-NET website [66].

**Table 3.** Ground motions.

| Station Name | Event Date | Distance | Ground Motion ID | PGA (m/s²) | |
|---|---|---|---|---|---|
| | | | | EW | NS |
| K-NET Kumamoto (KMM) | 14 April 2016 | 6 km | KMM0414 | 3.814 | 5.744 |
| | 16 April 2016 | 5 km | KMM0416 | 6.162 | 8.272 |
| K-NET Uto (UTO) | 14 April 2016 | 15 km | UTO0414 | 3.042 | 2.635 |
| | 16 April 2016 | 12 km | UTO0416 | 7.711 | 6.515 |
| KIK-NET Mashiki (MAS) | 14 April 2016 | 6 km | MAS0414 | 9.250 | 7.598 |
| | 16 April 2016 | 7 km | MAS0416 | 11.569 | 6.530 |

Figure 9 shows the recorded accelerations observed at the three stations. The present study uses the first 60 s of the as-recorded acceleration records shown in the figure for nonlinear time-history analysis.

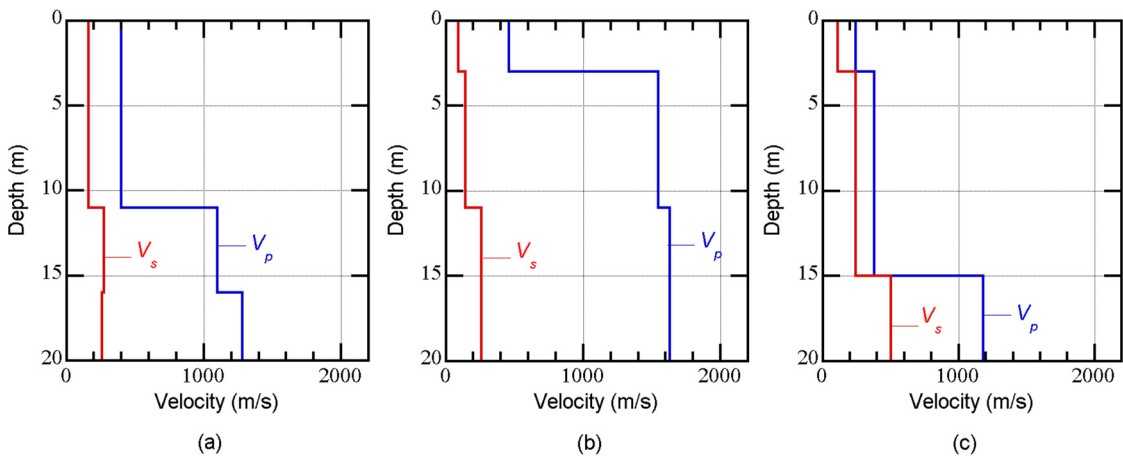

**Figure 8.** Primary and shear wave profiles: (**a**) KMM; (**b**) UTO; (**c**) MAS.

**Figure 9.** Ground motion time histories: (**a**) KMM; (**b**) UTO; (**c**) MAS.

Figure 10 shows the elastic pseudo-velocity spectra ($_pS_V$) of the accelerations. The viscous damping for the calculation of $_pS_V$ is set at 0.05. The "design" earthquake shown in

this figure is the code-specified spectrum (soil condition: type-2) of the Building Standard Law of Japan [77] defined as

$$_pS_V(T, 0.05) = \frac{T}{2\pi} {}_pS_A(T, 0.05), \tag{8}$$

$$_pS_A(T, 0.05) = \begin{cases} 4.8 + 45T \quad \text{m/s}^2 & T \leq 0.16 \text{ s} \\ 12.0 & 0.16 \text{ s} < T \leq 0.864 \text{ s} \\ 12.0(0.864/T) & T > 0.864 \text{ s} \end{cases} \tag{9}$$

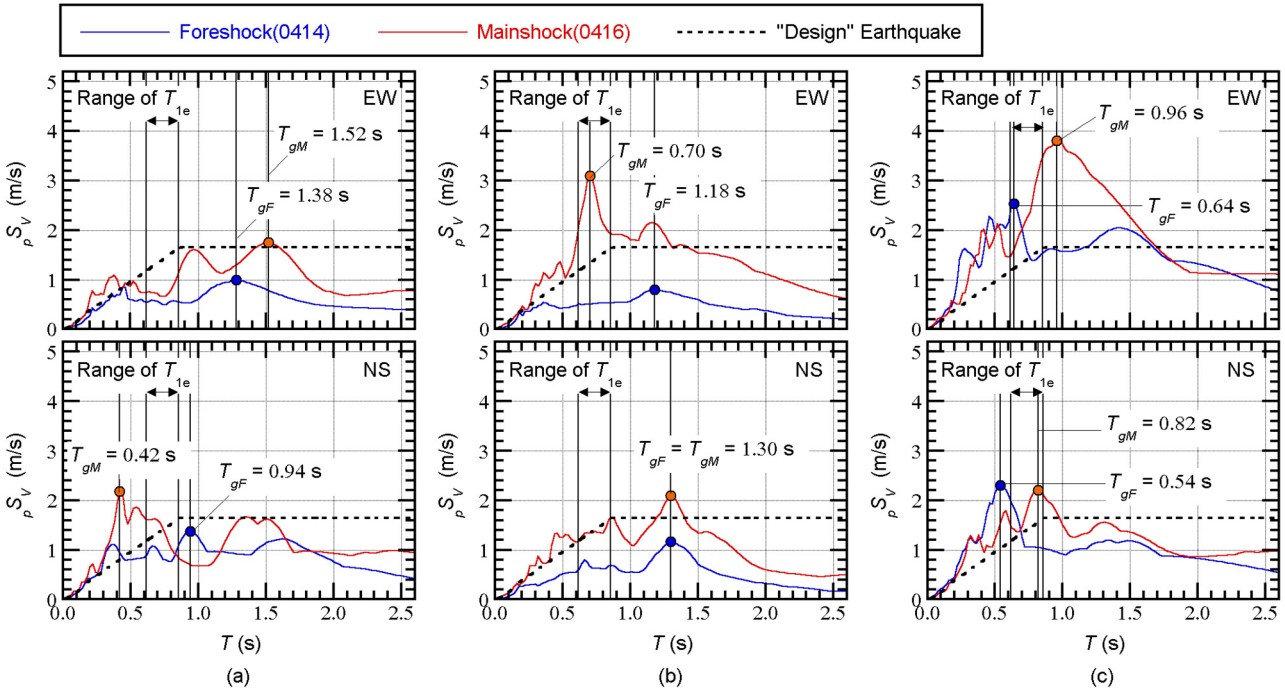

**Figure 10.** Comparisons of the pseudo-velocity spectrum of the observed ground motions (foreshock, aftershock) and the design earthquake: (**a**) KMM; (**b**) UTO; (**c**) MAS.

Note that the design earthquake spectrum is used for the design of models RCDC1 and RCDC2. The calculated spectra of the mainshock are close to the design earthquake spectrum at KMM (Figure 10a). In contrast, the calculated spectrum of the mainshock exceeds the design earthquake spectrum at MAS (Figure 10c). Note that the spectrum of the east–west (EW) component of the foreshock at MAS also exceeds the design earthquake spectrum, which implies that the EW components of both the foreshock and mainshock at MAS are more intense than the design earthquake.

Figure 10 also shows the predominant periods of the foreshock and mainshock, denoted $T_{gF}$ and $T_{gM}$, respectively. The predominant period is defined as the peak period of $_pS_V$, following Miranda [78] and Ruiz-García [14,17]. The figure reveals that the relation of the two predominant periods ($T_{gF}$ and $T_{gM}$) depends on the site and component. In the cases of the EW-component at KMM and both components at MAS, $T_{gM}$ is longer than $T_{gF}$, whereas $T_{gM}$ is shorter than $T_{gF}$ for the north–south (NS) component at KMM and the EW-component at UTO.

Table 4 lists the cases of ground motion considered in this study. Here, Cases F and M are, respectively, the single acceleration of only the foreshock and that of the mainshock, whereas cases FM and MF are sequential accelerations, with Case FM following the recorded order of first the foreshock (e.g., KMM0414EW) and second the mainshock (e.g., KMM0416EW) and Case MF following the opposite sequence of first the mainshock and second the foreshock. A time interval of 30 s is set between the first and second accelerations.

**Table 4.** Cases of ground motion.

| Case | Acceleration Sequence |
|------|----------------------|
| Case-F | Foreshock (0414) only |
| Case-FM | Foreshock (0414) + Mainshock (0416) |
| Case-M | Mainshock (0416) only |
| Case-MF | Mainshock (0416) + Foreshock (0414) |

## 3. Analysis Results

### 3.1. Peak Response

#### 3.1.1. Relative Displacement and Story Drift

Figure 11 compares the distribution of the peak relative displacement in the four cases for each model. The input ground motion set is the EW component at KMM (KMM-EW). The figure shows that the peak response in Case FM is larger than that in Case F, whereas the peak response in Case MF is the same as that in Case M for all models. Specifically, as shown in Figure 11a, the peak response of BareRC1 obtained in Case FM is larger than that obtained in Cases MF and M. This implies that, in Case FM, the damage to members due to the foreshock affects the response of BareRC1 during the mainshock (following the foreshock).

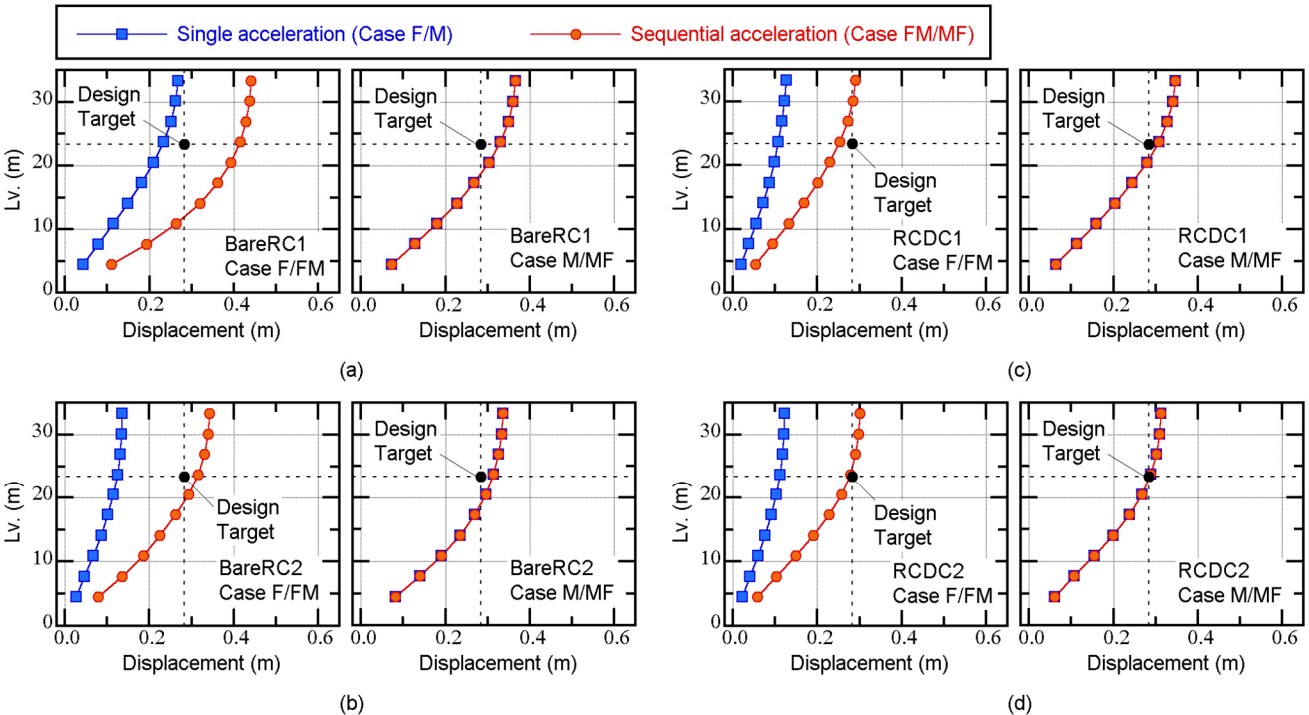

**Figure 11.** Distribution of the peak relative displacement (EW component of KMM-EW): (**a**) BareRC1; (**b**) BareRC2; (**c**) RCDC1; (**d**) RCDC2.

For the MRFs with steel damper columns, the peak response of RCDC1 (Figure 11c) in Case FM is within the design target, whereas it slightly exceeds the design target in Case MF. Similar results are obtained for RCDC2 as shown in Figure 11d. Figure 10a shows that the pseudo-velocity spectra of the foreshock and mainshock obtained from the EW components at KMM are close to those of the design earthquake. Figure 11c,d thus show that the strength demands of those two models are properly determined using the simplified design method proposed in the previous study [3].

Figure 12 compares the distribution of the peak relative displacement in the case that the input ground is set as MAS-EW. The figure shows that the peak displacement is larger than that in the case of KMM-EW shown in Figure 11. Specifically, the peak response of

BareRC1 in Case FM is the same as that in Case F, whereas the peak responses of the other models in Case FM are larger than those in Case F. Additionally, the peak responses of RCDC1 and RCDC2 exceed the design target in all four cases, as shown in Figure 12c,d. This result is consistent with the observation from Figure 10c that the pseudo-velocity spectra obtained from the EW components at MAS are larger than those of the design earthquake in a wide range of the period.

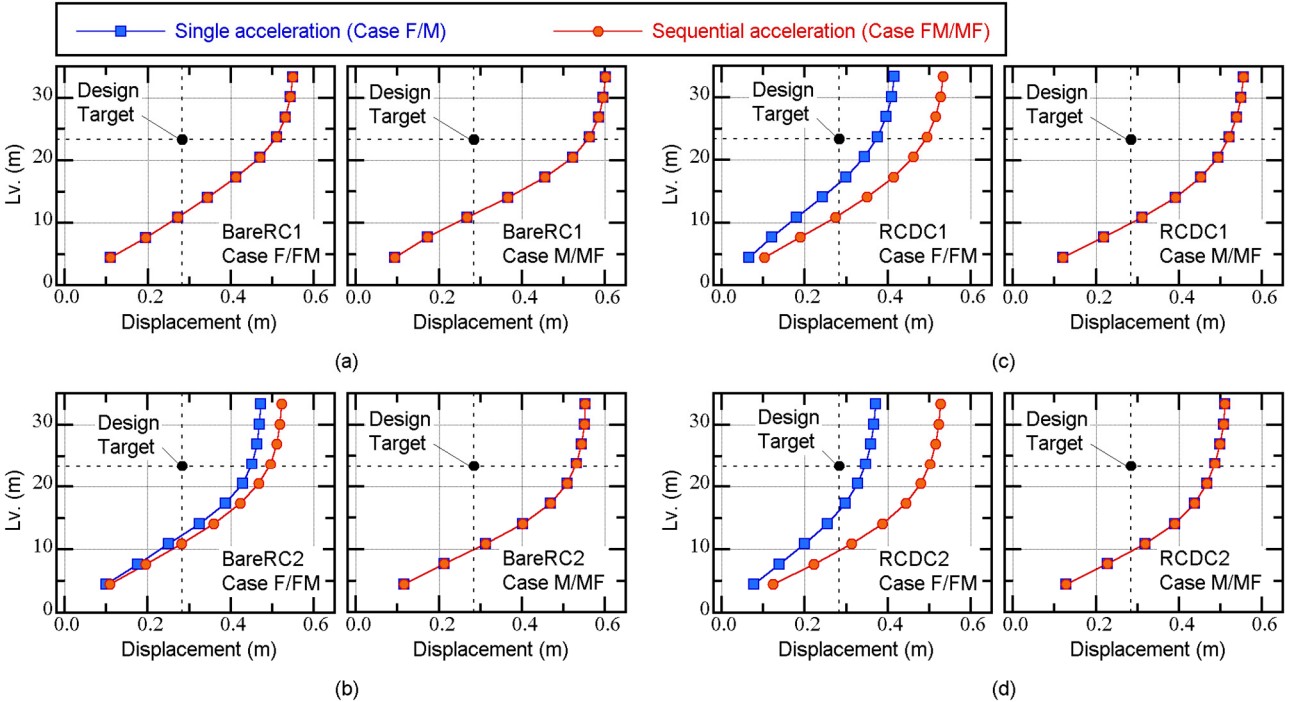

**Figure 12.** Distribution of the peak relative displacement (MAS-EW): (**a**) BareRC1; (**b**) BareRC2; (**c**) RCDC1; (**d**) RCDC2.

Figures 13 and 14 compare the peak story drift in the four cases for each model. The input ground motion sets are KMM-EW (Figure 13) and MAS-EW (Figure 14). The figures confirm the results in Figures 11 and 12. Note that the largest peak story drifts of RCDC1 and RCDC2 are close to 1/75 in the case of KMM-EW. The value of 1/75 is the assumed target drift in the design of RCDC1 and RCDC2, and the responses of both models under the sequential accelerations of KMM-EW (both Cases FM and MF) are close to the assumed design limit. In contrast, the responses of both RCDC1 and RCDC2 greatly exceed the assumed design limit in the case of MAS-EW.

Figure 15 compares the peak story drifts in the cases of single and sequential accelerations to clarify the effect of sequential accelerations on the peak story drift. It is seen that there is a notable difference in the peak drift between Cases F and FM. The difference in the peak drift between Cases F and FM is more pronounced for the MRFs with steel damper columns (RCDC1 and RCDC2) than for the MRFs without dampers (BareRC1 and BareRC2). In contrast, the difference in the peak drift between Cases M and MF is negligibly small for all models. These observations imply that the peak responses of all models under sequential accelerations studied here are governed by the mainshock. This point is discussed further in Section 4.

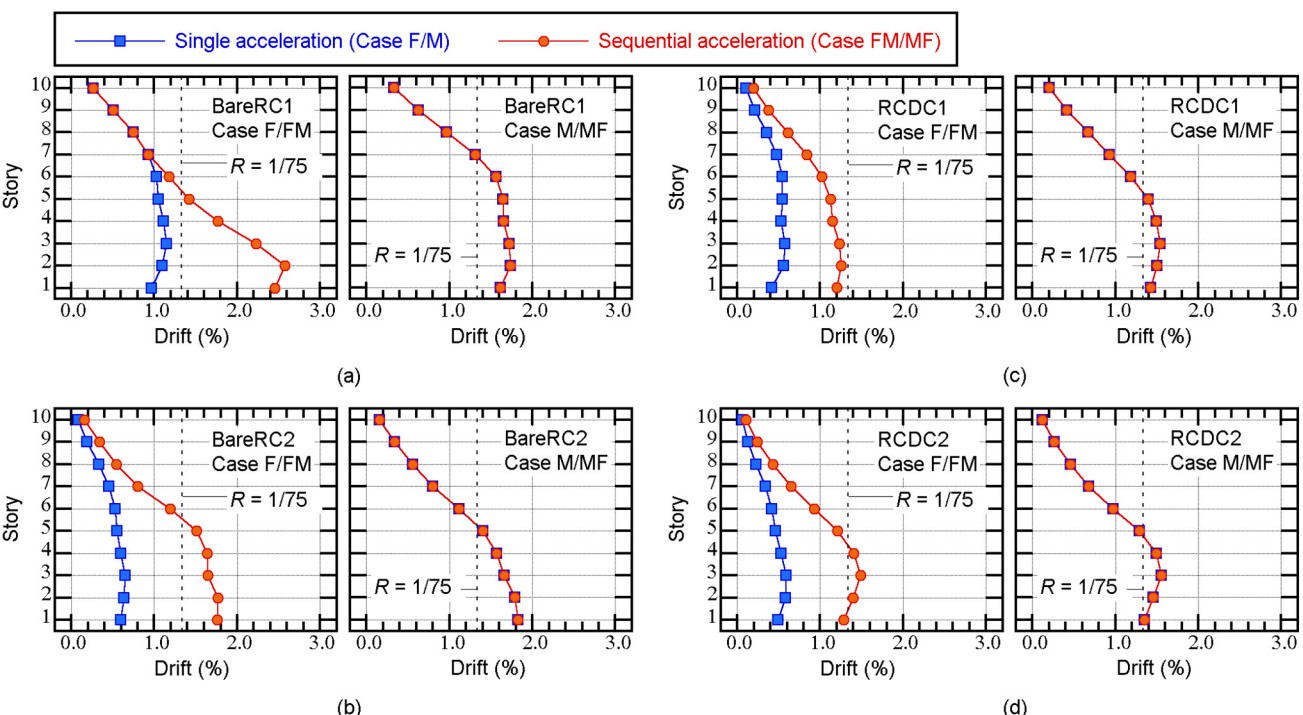

**Figure 13.** Distribution of the peak story drift (KMM-EW): (**a**) BareRC1; (**b**) BareRC2; (**c**) RCDC1; (**d**) RCDC2.

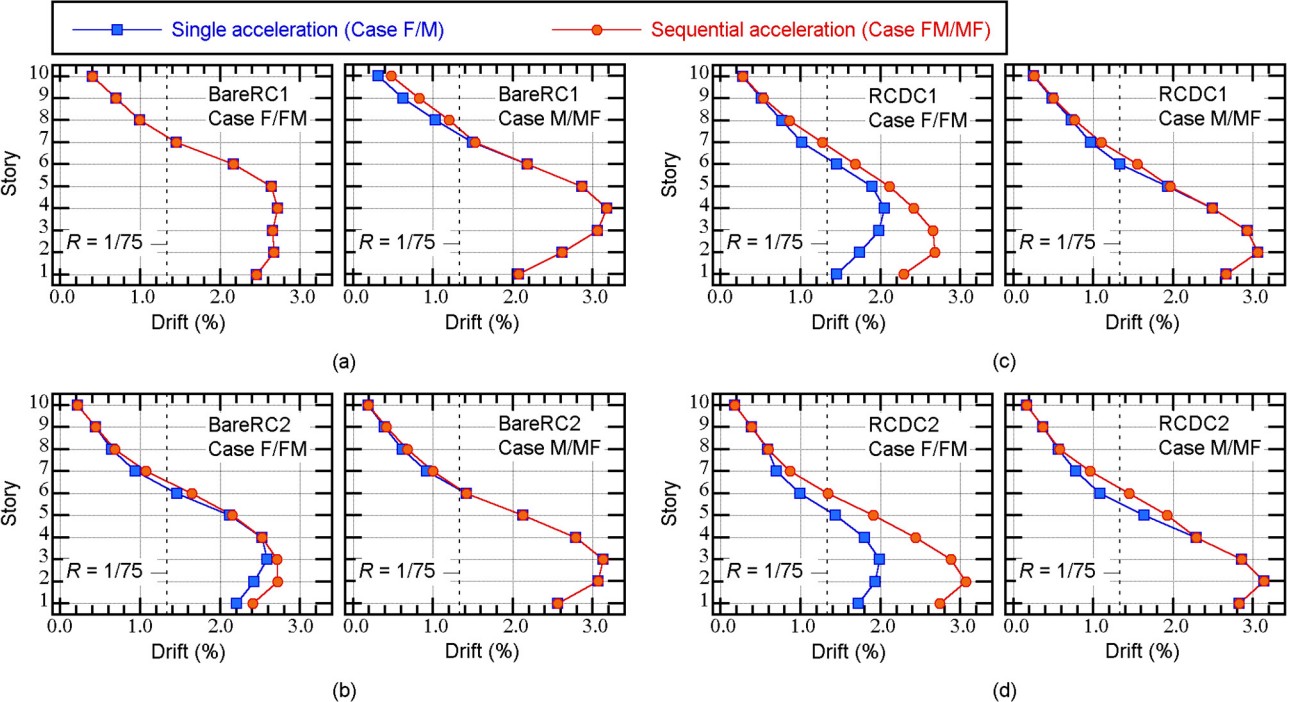

**Figure 14.** Distribution of the peak story drift (MAS-EW): (**a**) BareRC1; (**b**) BareRC2; (**c**) RCDC1; (**d**) RCDC2.

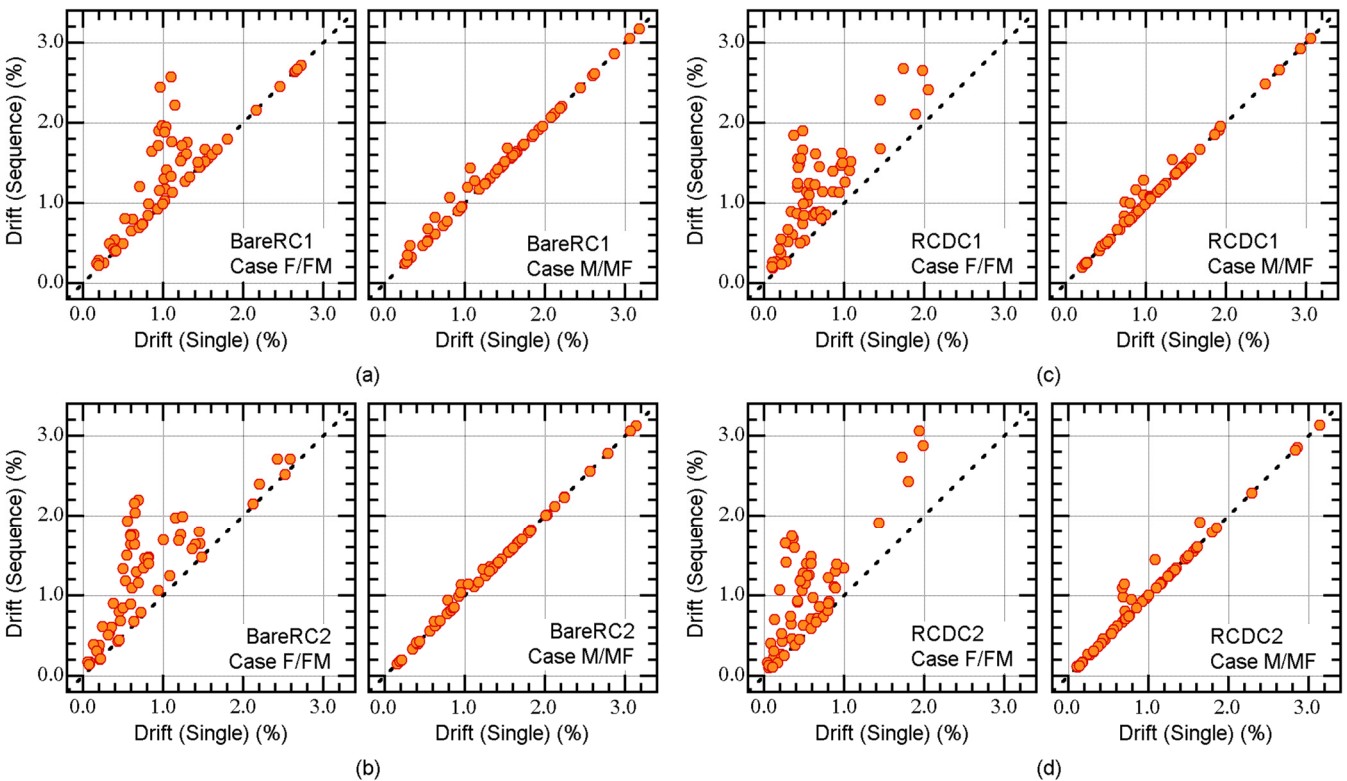

**Figure 15.** Comparisons of the peak story drift in the case of single and sequential acceleration: (**a**) BareRC1; (**b**) BareRC2; (**c**) RCDC1; (**d**) RCDC2.

### 3.1.2. Member Deformation

Next, the peak responses of the member deformation (plastic rotation of the RC beam end and shear strain of the damper panel) are investigated. Figure 16 compares the peak rotation of the plastic hinge at the beam end. MAS-EW is used as the input ground motion set because it gives the largest response among all input ground motion sets. The figure shows the plastic hinge rotation at beam ends in the inner frame (i.e., the right-side ends of beam $X_2X_3$ in Frame $Y_3$ for BareRC1 and RCDC1 and the left-side ends of beam $X_3X_4$ in Frame $Y_3$ for BareRC2 and RCDC2). It is seen that the peak plastic rotation in Case FM is larger than that in Case F except for the model BareRC1, whereas the plastic rotation in Case MF is the same as that in Case M. This result is consistent with Figure 14. Figure 16 also shows that the plastic rotation of beams at upper floor levels (higher than level $Z_7$) is zero; there is no yielding of these beams.

Figure 17 compares the peak plastic hinge rotation at the beam end in the cases of single and sequential acceleration. There is a notable difference in the peak plastic rotation between Cases F and FM, whereas the difference in the peak drift between Cases M and MF is negligibly small for all models. This result is consistent with Figure 15.

Figure 18 compares the peak shear strain of the damper panel in RCDC1 and RCDC2, where the input ground motion set is MAS-EW. It is seen that the peak shear strain of the damper panel is greater for the lower stories. In addition, the peak shear strain in Case FM is greater than that in Case F, whereas the peak shear strain in Case MF is similar to that in Case M.

Figure 19 compares the peak shear strain of the damper panel in cases of single and sequential acceleration. There is a notable difference in the peak shear strain between Cases F and FM, whereas the difference in the peak drift between Cases M and MF is negligibly small for both RCDC1 and RCDC2.

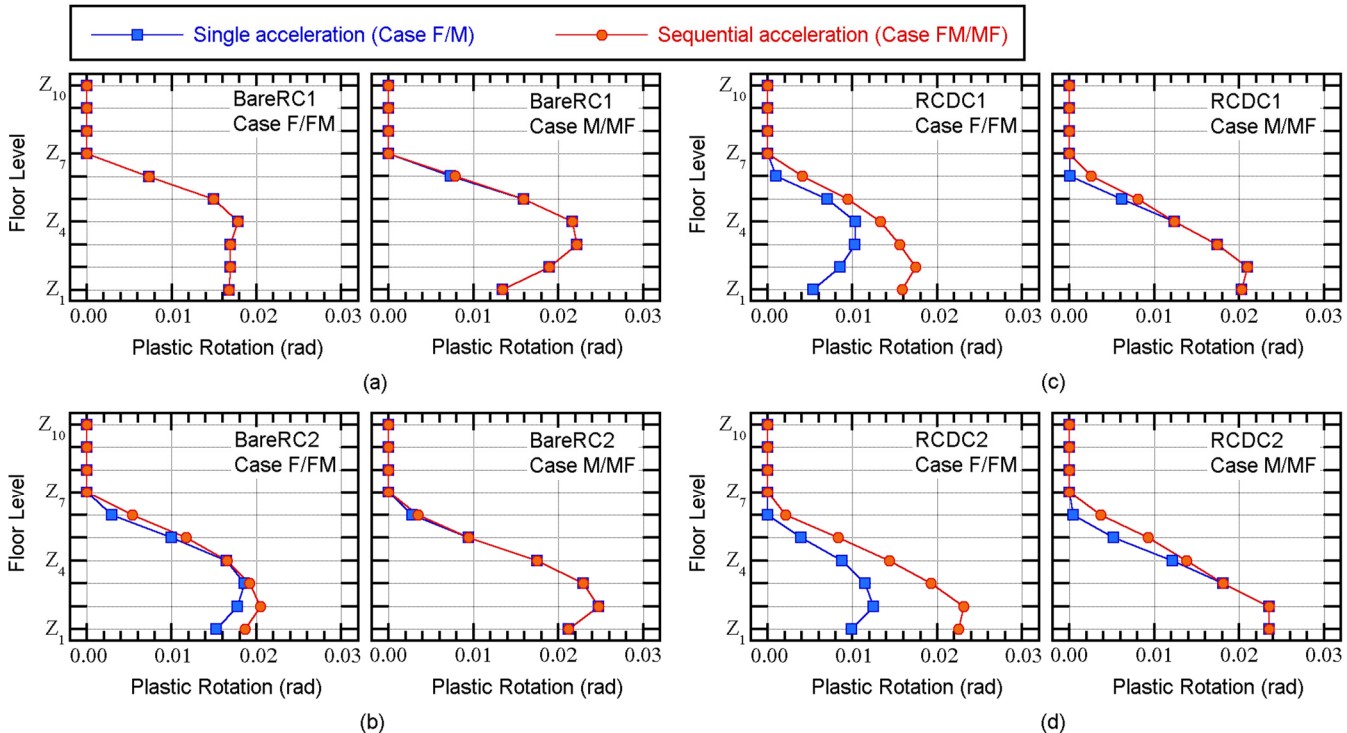

**Figure 16.** Distribution of the peak plastic hinge rotation at the beam end (MAS-EW): (**a**) BareRC1; (**b**) BareRC2; (**c**) RCDC1; (**d**) RCDC2.

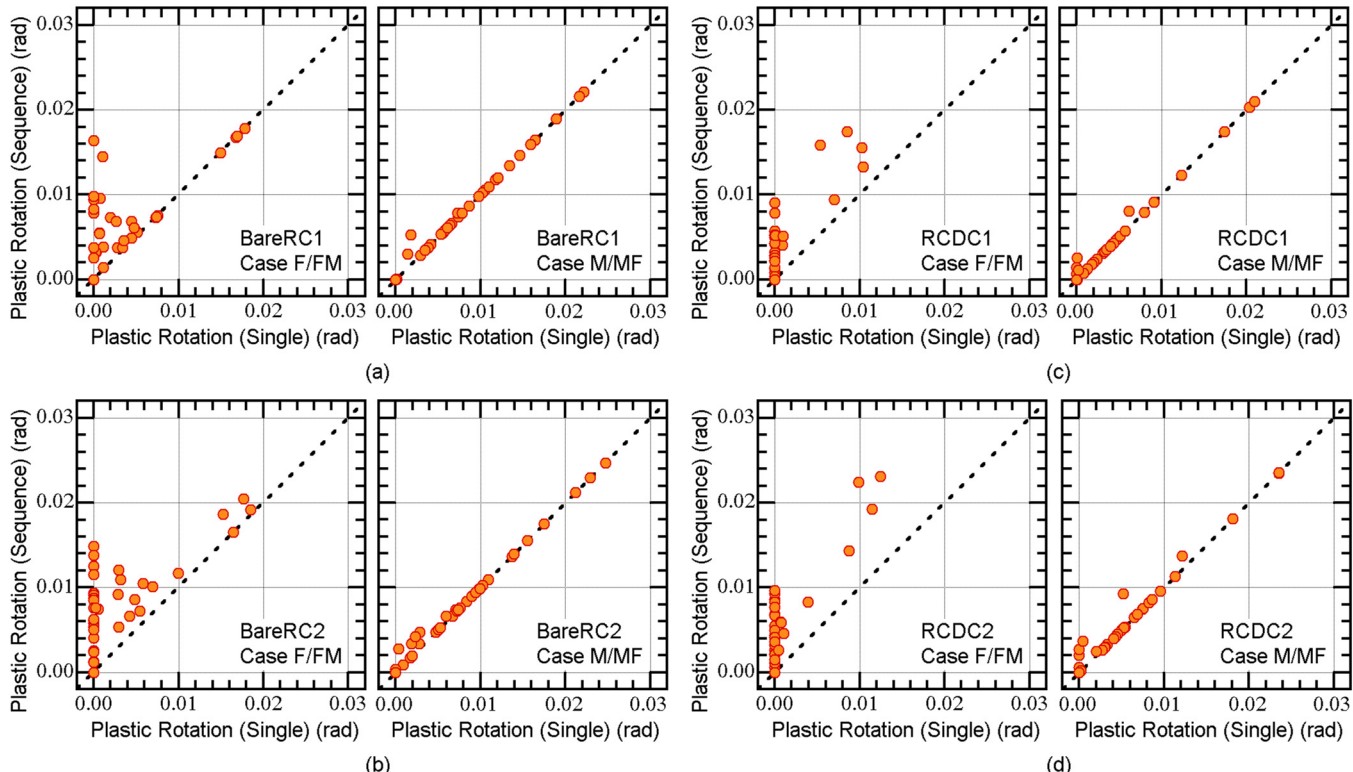

**Figure 17.** Comparisons of the peak plastic hinge rotation at the beam end in cases of single and sequential acceleration: (**a**) BareRC1; (**b**) BareRC2; (**c**) RCDC1; (**d**) RCDC2.

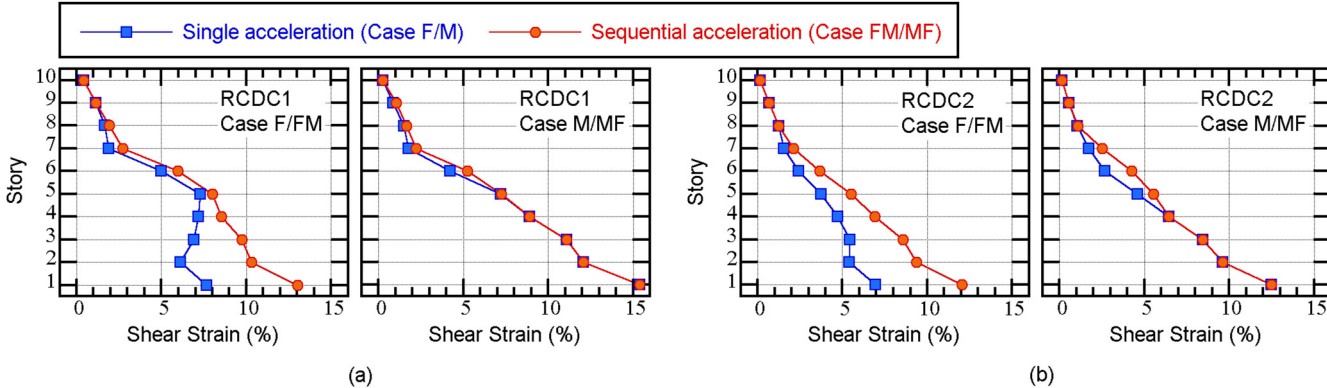

**Figure 18.** Distribution of the peak shear strain of the damper panel (MAS-EW): (**a**) RCDC1; (**b**) RCDC2.

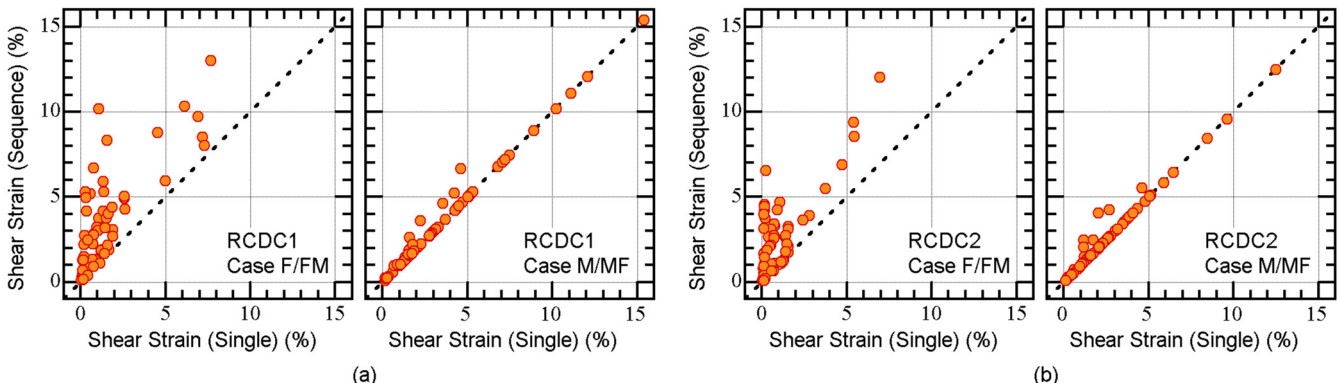

**Figure 19.** Comparisons of the peak shear strain of the damper panel in the cases of single and sequential acceleration: (**a**) RCDC1; (**b**) RCDC2.

### *3.2. Cumulative Response*

This subsection discusses the cumulative responses of the four building models. First, the cumulative responses of the overall building (i.e., cumulative input energy $E_I$, cumulative strain of the RC MRF and steel damper columns, $E_{Sf}$ and $E_{Sd}$, respectively, and cumulative viscous damping energy $E_D$) are discussed. The cumulative strain energy of the member (plastic hinge of the RC beam and damper panel) is then discussed.

#### 3.2.1. Cumulative Response of the Overall Building

Figure 20 shows the cumulative strain energy of the RC MRF and steel damper columns and the cumulative viscous damping energy per unit mass. Note that the total input energy $E_I$ is expressed as the sum of $E_{Sf}$, $E_{Sd}$, and $E_D$.

The following observations are made for Figure 20.

- The total input energy $E_I$ of the sequential accelerations is greater than that of the single acceleration: e.g., $E_I$ in Case FM is greater than that in Case F.
- In most cases for BareRC1 and BareRC2, $E_I$ is mostly absorbed as cumulative strain energy of the RC MRF, $E_{Sf}$.
- In most cases for RCDC1 and RCDC2, the total input energy $E_I$ is greater than that of BareRC1 and BareRC2. However, a large proportion of $E_I$ is absorbed as the cumulative strain energy of the steel damper columns, $E_{Sd}$. The relative amounts of $E_{Sf}$ and $E_{Sd}$ depend on the model and analysis case.

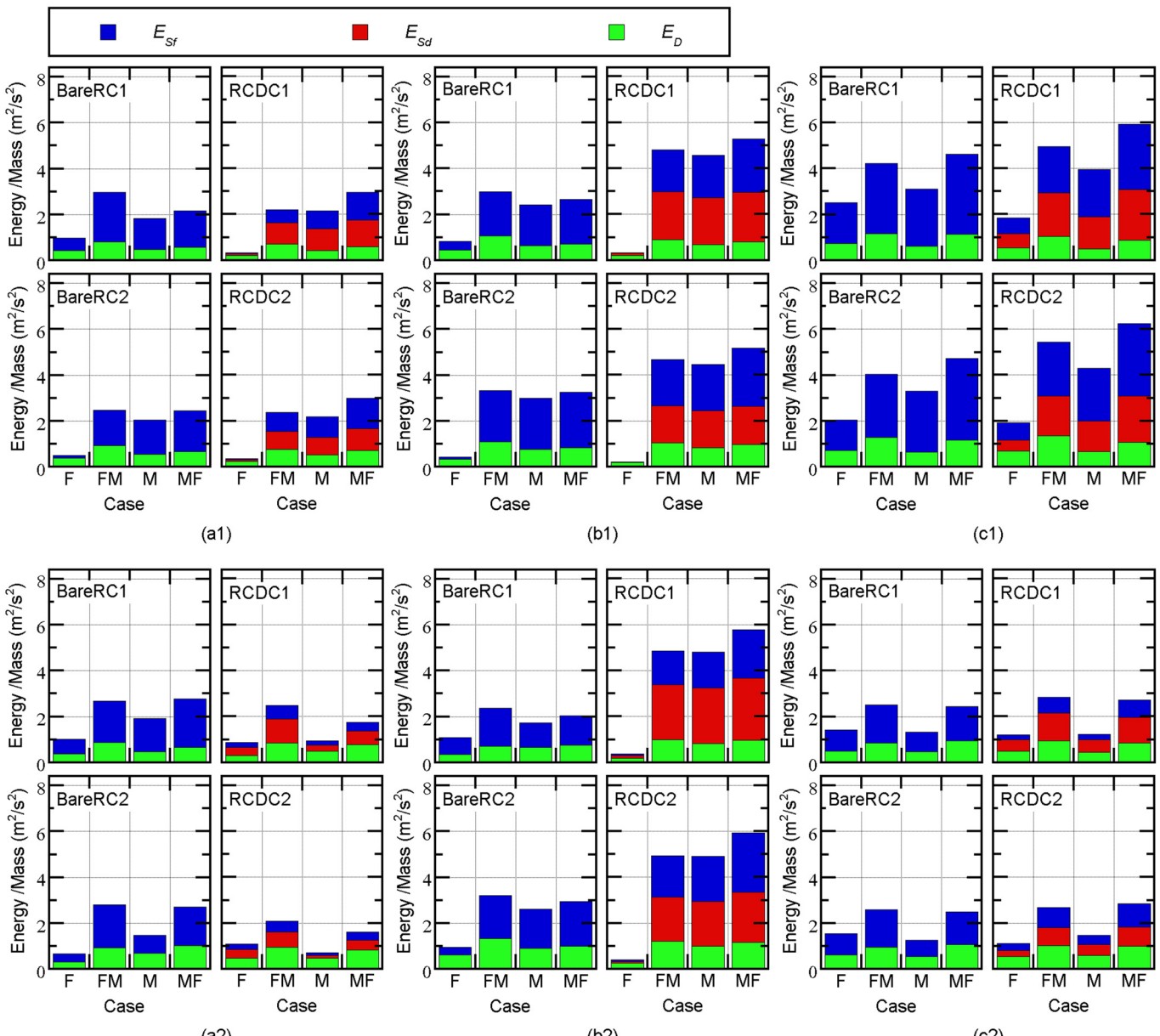

**Figure 20.** Comparisons of the cumulative energy per unit mass of the overall building in the cases of single acceleration and sequential acceleration: (**a1**) KMM-EW; (**a2**) KMM-NS; (**b1**) UTO-EW; (**b2**) UTO-NS; (**c1**) MAS-EW; (**c2**) MAS-NS.

### 3.2.2. Cumulative Strain Energy of a Member

First, the cumulative strain energy of the plastic hinges at the RC beam end is investigated. The normalized strain energy of the $k$th plastic hinge, $NE_{Sfk}$, is defined as

$$NE_{Sfk} = \frac{E_{Sfk}}{M_{yk}\theta_{yk}} = \frac{1}{M_{yk}\theta_{yk}} \int_0^{t_d} M_k(t)\dot{\theta}_k(t)dt, \tag{10}$$

where $M_k(t)$ and $\theta_k(t)$ are, respectively, the moment and rotation of the $k$th plastic hinge at time $t$, and $t_d$ is the time length of the nonlinear time-history analysis.

Figure 21 compares $NE_{Sfk}$ in the cases of single and sequential accelerations for BareRC1 and BareRC2. In both models, $NE_{Sfk}$ at beam ends in the inner frame (the same beam ends considered in Section 3.1.2) is investigated. The figure shows that $NE_{Hfk}$ is

notably greater in Case FM than in Case F. In addition, the increase in $NE_{Sfk}$ from Case M to Case MF is not negligible. This implies that the effect of sequential accelerations on the cumulative strain energy is more pronounced than that on the peak response. The figure also shows that, in general, $NE_{Sfk}$ is larger for BareRC2 than for BareRC1. This is because the yield deformation angle of beams ($\theta_{yk}$) is smaller for BareRC2 than for BareRC1.

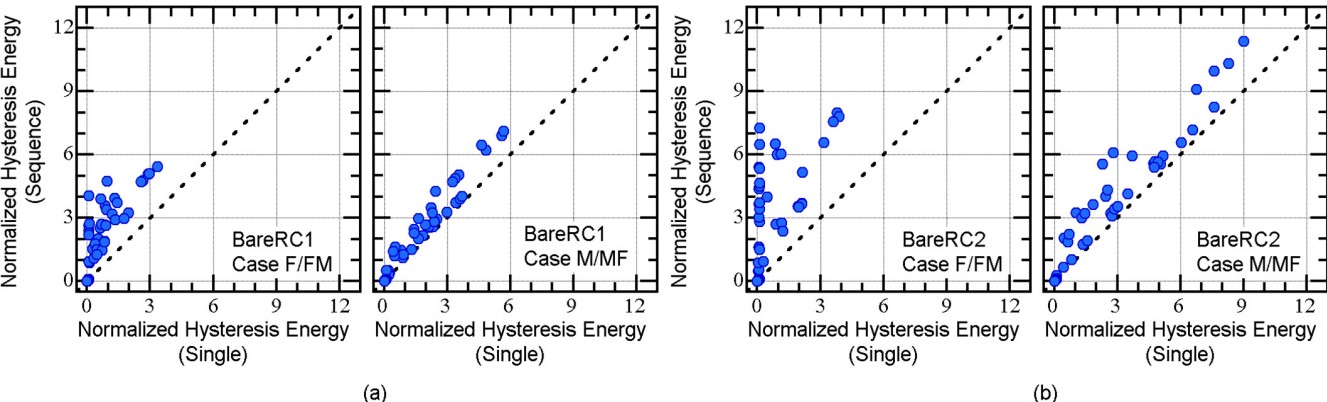

**Figure 21.** Comparisons of the cumulative strain energy of the plastic hinges at the beam end in the cases of single and sequential acceleration: (**a**) BareRC1; (**b**) BareRC2.

Figure 22 compares $NE_{Sfk}$ in the cases of single and sequential accelerations for RCDC1 and RCDC2. For RCDC1 and RCDC2, $NE_{Sfk}$ at the long-span beam ends (the same beam ends considered in Section 3.1.2) and short-span beam ends (the right-side ends of beam $X_2X_3$ in Frame $Y_2$ for RCDC1 and the left-side ends of beam $X_3X_4$ in Frame $Y_2$ for RCDC2) is investigated.

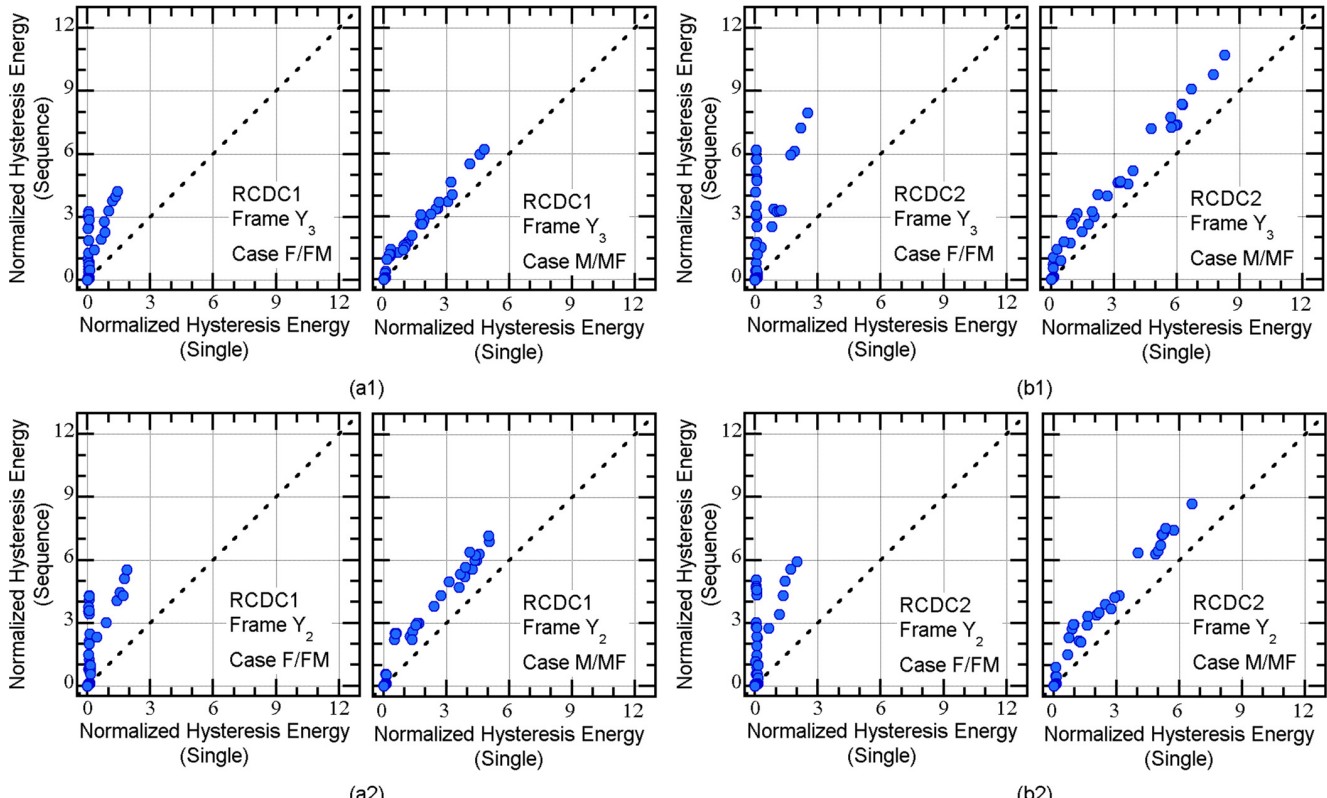

**Figure 22.** Comparisons of the cumulative strain energy of the plastic hinge at the beam end for single and sequential accelerations: (**a1**) RCDC1 (Frame $Y_3$); (**a2**) RCDC1 (Frame $Y_2$); (**b1**) RCDC2 (Frame $Y_3$); (**b2**) RCDC2 (Frame $Y_2$).

The figure shows a notable increase in $NE_{Sfk}$ due to sequential accelerations for RCDC1 and RCDC2. Specifically, for RCDC1, the increase in $NE_{Sfk}$ in Frame $Y_2$ is larger than that in Frame $Y_3$. The reason is that (i) the yield deformation angle of beams ($\theta_{yk}$) in Frame $Y_2$ is smaller than that in Frame $Y_3$ and (ii) the cyclic stiffness degradation is more pronounced for the short-span beam (Frame $Y_2$) than for the longer span beam (Frame $Y_3$).

Next, the cumulative strain energy of the damper panel is investigated. The normalized strain energy of the $k$th damper panel, $NE_{Sdk}$, is defined as

$$NE_{Sdk} = \frac{E_{Sdk}}{Q_{yDLk}\gamma_{yDLk}h_{d0k}} = \frac{1}{Q_{yDLk}\gamma_{yDLk}}\int_{0}^{t_d} Q_{Dk}(t)\dot{\gamma}_{Dk}(t)dt, \qquad (11)$$

where $Q_{Dk}(t)$ and $\gamma_{Dk}(t)$ are, respectively, the shear force and shear deformation angle of the $k$th damper panel at time $t$.

Figure 23 compares $NE_{Sdk}$ in the cases of single and sequential acceleration. It is seen that the increase in $NE_{Sdk}$ due to the sequential accelerations is pronounced in both models. In addition, the value of $NE_{Sdk}$ for RCDC1 is greater than that for RCDC2.

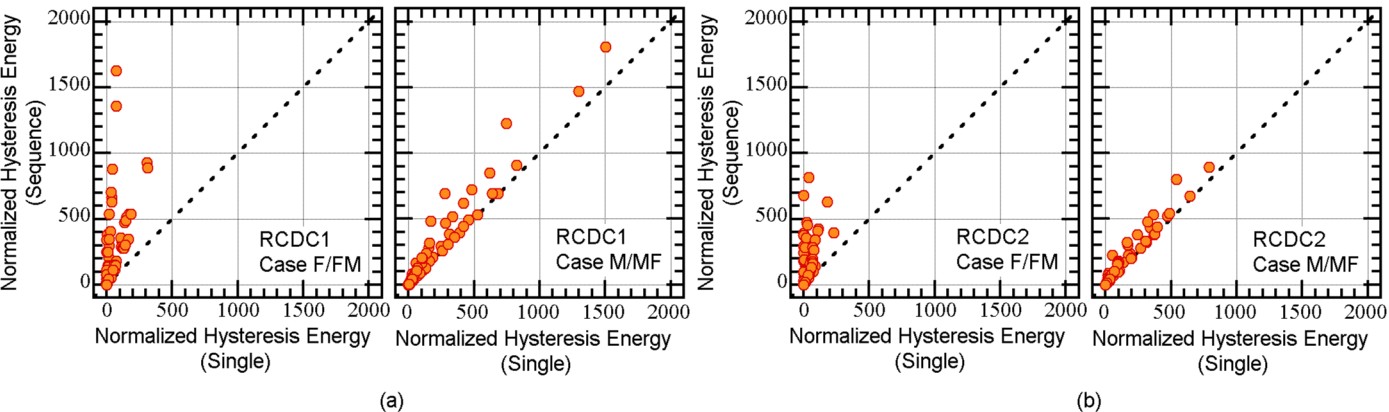

**Figure 23.** Comparisons of the cumulative strain energy of the damper panel in the cases of single and sequential acceleration: (**a**) RCDC1; (**b**) RCDC2.

It is concluded from the results presented in this subsection that the effect of the sequential accelerations on the cumulative strain energy in the member is not negligible. In contrast with the peak deformation, the cumulative strain energy accumulates in the event of sequential accelerations.

### 3.3. Effectiveness of Steel Damper Columns in Reducing the Seismic Response

In this subsection, the effectiveness of the steel damper column in reducing the peak and cumulative responses of the RC members is discussed by comparing results for MRFs without dampers (BareRC1, BareRC2) and those with steel damper columns (RCDC1, RCDC2).

Figure 24 compares the peak story drift for models without and with steel damper columns. As shown in Figure 24a, in general, the drift for RCDC1 is smaller than that for BareRC1. The effect of the steel damper columns in the reduction of the peak drift is obvious in the case of sequential accelerations as well as the case of single acceleration. A similar observation can be made in comparing RCDC2 and BareRC2.

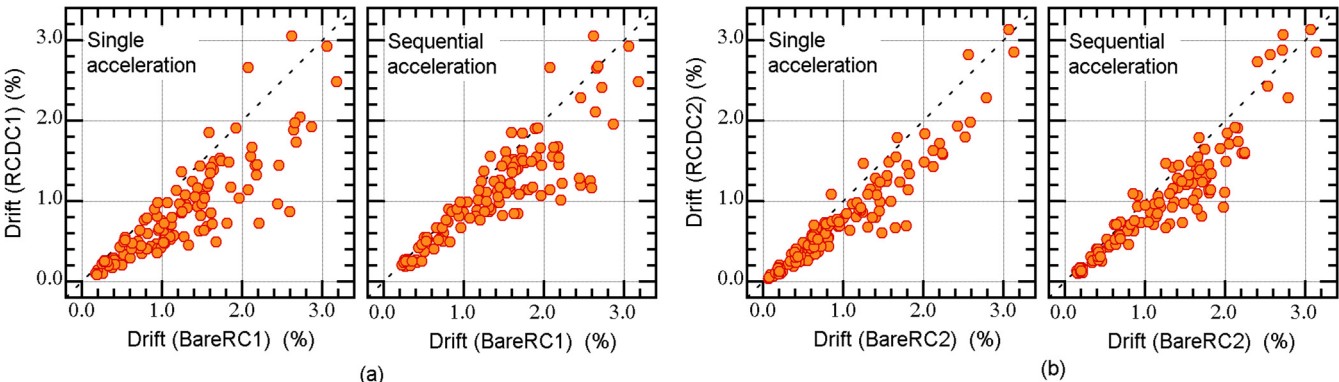

**Figure 24.** Comparisons of the peak story drift for models without and with damper columns: (**a**) BareRC1 and RCDC1; (**b**) BareRC2 and RCDC2.

Figure 25 compares the peak plastic hinge rotation at the beam ends for Frame Y$_3$. It is seen that the peak plastic rotation of the models with steel damper columns (RCDC1, RCDC2) is smaller than that of the models without dampers (BareRC1, BareRC2). It is concluded from the comparisons that the steel damper column is effective for the reduction in the peak response of the RC MRF, as far as the models studied herein are concerned.

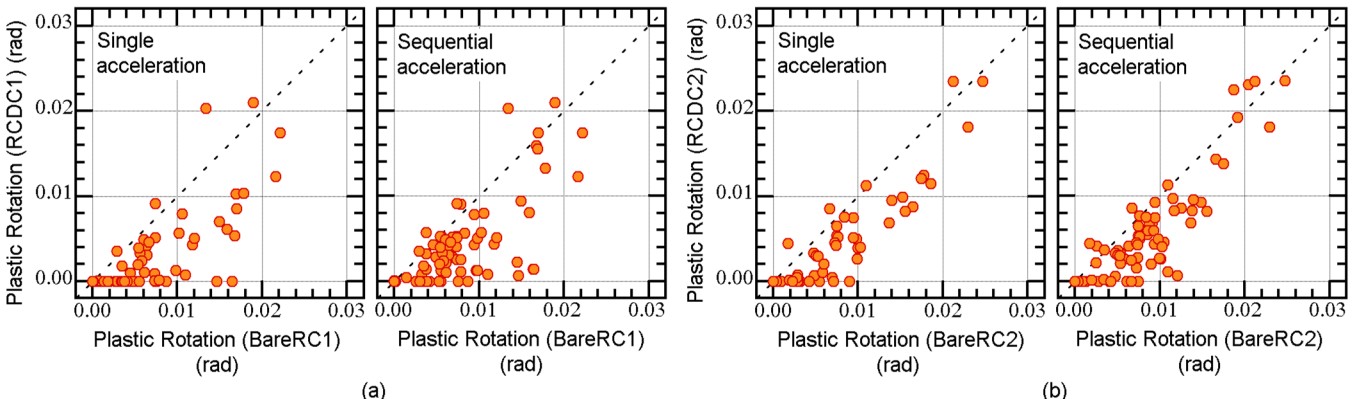

**Figure 25.** Comparisons of the peak plastic rotation at beam ends in models without and with damper columns: (**a**) BareRC1 and RCDC1; (**b**) BareRC2 and RCDC2.

The cumulative responses are next compared. Figure 26 compares the cumulative strain energy of the RC MRF per unit mass. As shown in Figure 26a, in most cases, the cumulative strain energy of RCDC1 is less than that of BareRC1 for both single acceleration and sequential acceleration. Similar observations are made for Figure 26b.

Figure 27 compares the normalized cumulative strain energy at the beam ends ($NE_{Sfk}$). It is seen that $NE_{Sfk}$ with steel damper columns (RCDC1, RCDC2) is smaller than that without dampers (BareRC1, BareRC2). It is concluded from the comparisons that the steel damper column is effective for the reduction of the cumulative response of the RC MRF, as far as the models studied herein are concerned. Note that the total input energy of MRFs with steel damper columns is greater than that of MRFs without dampers in some cases, as shown in Figure 20. Even in such cases, those steel damper columns installed in the RC MRF play an important role of absorbing the seismic energy, as shown in Figure 27.

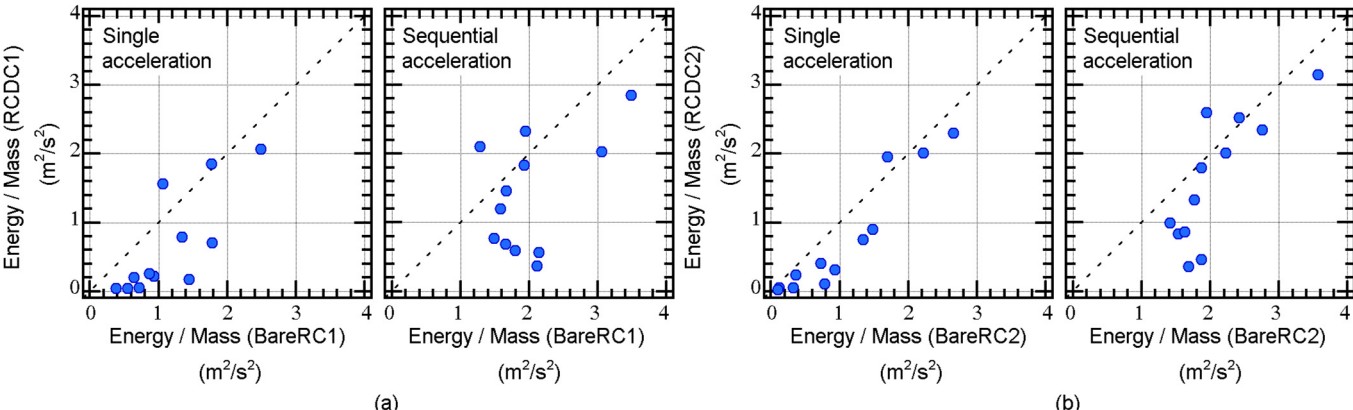

**Figure 26.** Comparisons of the cumulative strain energy of RC MRF for models without and with damper columns: (**a**) BareRC1 and RCDC1; (**b**) BareRC2 and RCDC2.

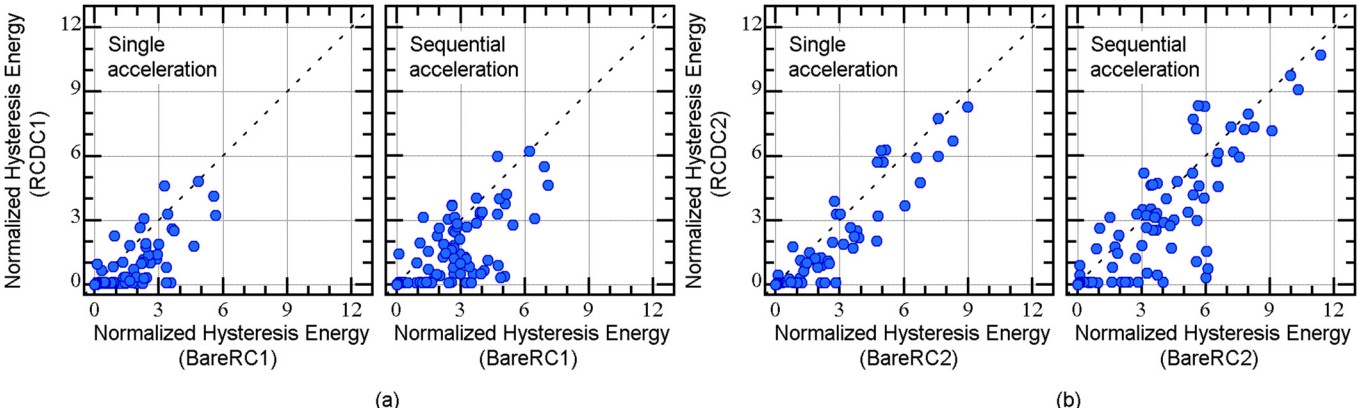

**Figure 27.** Comparisons of the cumulative strain energy at the plastic hinge at the beam end for models without and with damper columns: (**a**) BareRC1 and RCDC1; (**b**) BareRC2 and RCDC2.

### *3.4. Summary of Results*

The results presented in this section are summarized as follows.

- The peak responses of all models under sequential accelerations studied here are governed by the mainshock. The peak responses of all models in Case FM are notably greater than those in Case F. In contrast, the difference in the peak response between Case MF and Case M is limited.
- The effect of the sequential accelerations on the cumulative strain energy is not negligible. Unlike the peak response, the cumulative strain energy accumulates in the event of sequential accelerations.
- The steel damper column is effective in reducing the peak and cumulative responses of RC members, irrespective of whether a single acceleration or sequential accelerations is considered as the seismic input.

### 4. Evaluation of the First Modal Response

### *4.1. Method of Calculating the First Modal Response from the Results of Time-History Analysis*

This section calculates the equivalent displacement $D_1{}^*(t)$ and acceleration $A_1{}^*(t)$ from the results of nonlinear time-history analysis. This study assumes that the building model considered here oscillates predominantly in the first mode, and the first mode vector at the peak response ($_{peak}\Gamma_1{}_{\mathbf{peak}}\boldsymbol{\varphi}_1$) is then assumed from the relative horizontal displacement vector ($\mathbf{d}(t)$). Note that in other studies (e.g., [65]), the first mode vector is assumed by referring to pushover analysis considering the change in the first mode vector at each nonlinear stage (displacement-based mode-adaptive pushover analysis [76]). However,

because the present study considers the cyclic stiffness degradation of RC members and the strain-hardening effect of damper panels, the assumption of the first mode vector at the peak response from the pushover analysis results cannot be applied. Therefore, in this study, the assumption of $_{peak}\Gamma_1{}_{\mathbf{peak}}\boldsymbol{\varphi}_1$ is based on the results of nonlinear time-history analysis. The calculation method is described below.

### 4.1.1. Step 1: Determination of the "Peak Response Point"

The displacement at "the center of the mass" $D^*(t)$ is calculated as

$$D^*(t) = \frac{\mathbf{1^T M d}(t)}{\mathbf{1^T M 1}}.$$ (12)

The time $t_{peak}$ at which the absolute value of $D^*(t)$ is a maximum is then found.

### 4.1.2. Step 2: Determination of the First Mode Vector at the Peak Response Point

The first mode vector at the peak response point $(_{peak}\Gamma_1{}_{\mathbf{peak}}\boldsymbol{\varphi}_1)$ is assumed to be proportional to the relative horizontal displacement vector at time $t_{peak}$ $(\mathbf{d}\left(t_{peak}\right))$:

$$_{peak}\Gamma_1{}_{\mathbf{peak}}\boldsymbol{\varphi}_1 \approx \frac{\mathbf{d}\left(t_{peak}\right)^{\mathbf{T}}\mathbf{M 1}}{\mathbf{d}\left(t_{peak}\right)^{\mathbf{T}}\mathbf{M d}\left(t_{peak}\right)}\mathbf{d}\left(t_{peak}\right).$$ (13)

The effective first modal mass $(_{peak}M_1{}^*)$ is then calculated as

$$_{peak}M_1{}^* = {}_{peak}\Gamma_1{}_{\mathbf{peak}}\boldsymbol{\varphi}_1{}^{\mathbf{T}}\mathbf{M 1}.$$ (14)

### 4.1.3. Step 3: Calculation of the Equivalent Displacement and Acceleration of the First Modal Response

The equivalent displacement $D_1{}^*(t)$ and acceleration $A_1{}^*(t)$ are calculated as

$$D_1{}^*(t) = \frac{_{peak}\Gamma_1{}_{\mathbf{peak}}\boldsymbol{\varphi}_1{}^{\mathbf{T}}\mathbf{M d}(t)}{_{peak}M_1{}^*},$$ (15)

$$A_1{}^*(t) = \frac{_{peak}\Gamma_1{}_{\mathbf{peak}}\boldsymbol{\varphi}_1{}^{\mathbf{T}}\mathbf{f_R}(t)}{_{peak}M_1{}^*}.$$ (16)

In Equation (16), $\mathbf{f_R}(t)$ is the restoring force vector at time $t$.

### 4.1.4. Step 4: Calculation of the Momentary Input Energy and Hysteretic Dissipated Energy in a Half Cycle of the First Modal Response

The momentary input energy of the first modal response per unit mass $(\Delta E_1{}^*/M_1{}^*)$ is calculated from the time-derivative of the equivalent displacement $(\dot{D}_1{}^*(t))$ and the ground acceleration $(a_g(t))$ as

$$\frac{\Delta E_1{}^*}{M_1{}^*} = -\int\limits_{t}^{t+\Delta t} a_g(t)\dot{D}_1{}^*(t)dt.$$ (17)

In Equation (17), $t$ and $t + \Delta t$ are, respectively, the beginning and ending times of a half cycle, following the definition of Inoue and his group [59–61]. The maximum momentary input energy of the first modal response per unit mass $(\Delta E_1{}^*{}_{\max}/M_1{}^*)$ is defined as the maximum value of $\Delta E_1{}^*/M_1{}^*$ over the course of the whole seismic input.

Similarly, the hysteretic dissipated energy in a half cycle per unit mass ($\Delta E_{H1}{}^*/M_1{}^*$) is calculated as

$$\frac{\Delta E_{H1}{}^*}{M_1{}^*} = \int\limits_{t}^{t+\Delta t} A_1{}^*(t)\dot{D}_1{}^*(t)dt. \tag{18}$$

The hysteresis dissipated energy per unit mass in a half cycle at the maximum momentary energy input ($\Delta E_{H1}{}^*{}_{max}/M_1{}^*$) is defined as $\Delta E_{H1}{}^*/M_1{}^*$ in the half cycle that $\Delta E_1{}^*{}_{max}/M_1{}^*$ occurs.

For the convenience of discussion, the equivalent velocities of $\Delta E_1{}^*{}_{max}/M_1{}^*$ and $\Delta E_{H1}{}^*{}_{max}/M_1{}^*$ are defined as

$$V_{\Delta E1}{}^* = \sqrt{2\Delta E_1{}^*{}_{max}/M_1{}^*},\, V_{\Delta EH1}{}^* = \sqrt{2\Delta E_{H1}{}^*{}_{max}/M_1{}^*}. \tag{19}$$

### 4.2. Calculation Results

Figure 28 shows the time history of the first modal response for RCDC1 subjected to Uto-NS (Case FM). It is seen that the peak response occurs during UTO0416NS (the second earthquake). The momentary energy input is a maximum in the half cycle from $t = 118.60$ s to $t + \Delta t = 119.30$ s.

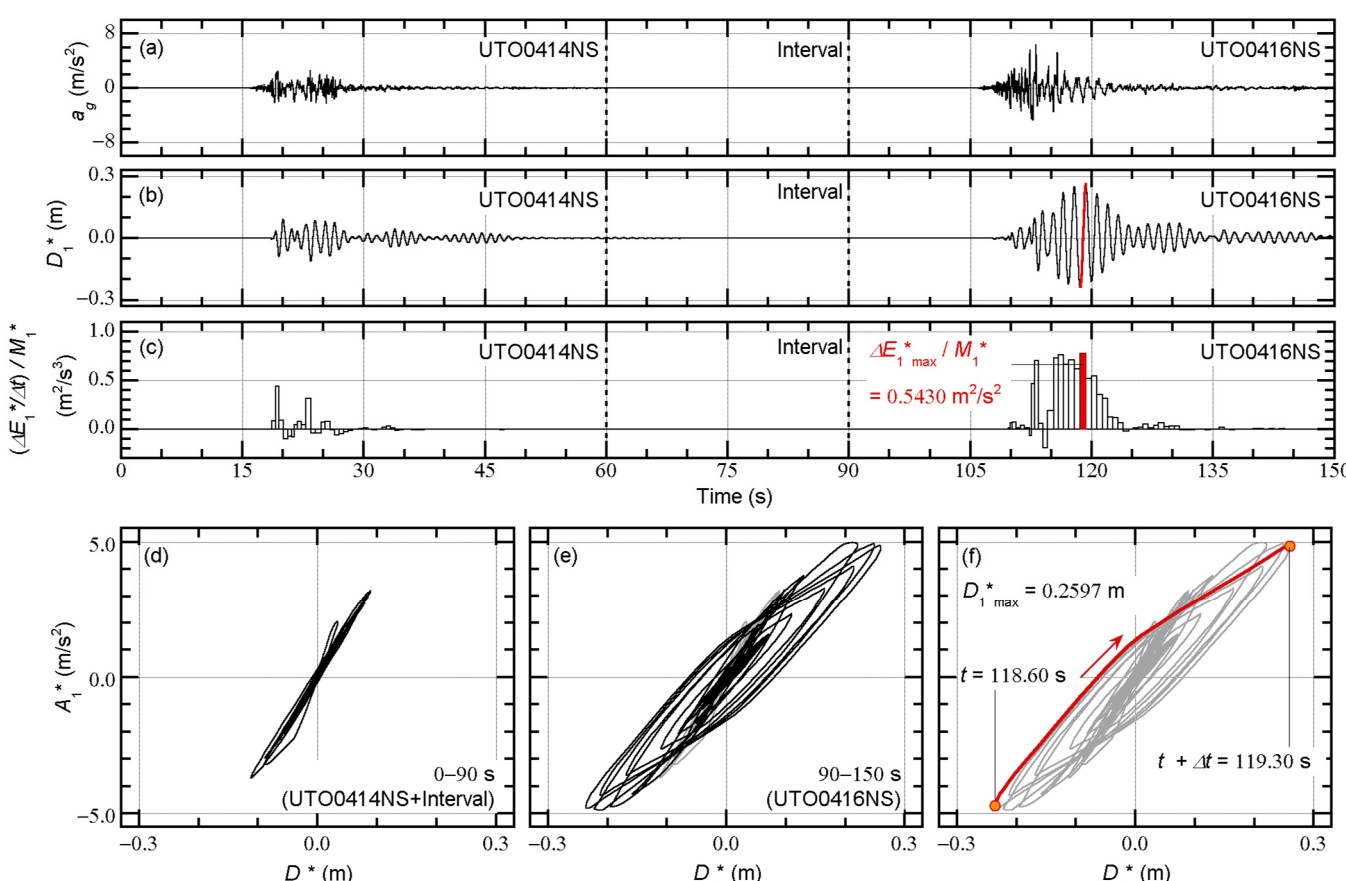

**Figure 28.** Calculated first modal response of RCDC1 subjected to UTO-NS (Case FM): (**a**) time history of the ground acceleration; (**b**) time history of the equivalent displacement; (**c**) time history of the momentary energy input per unit mass; (**d**) hysteresis loop of the $A_1{}^*$–$D_1{}^*$ relationship (0–90 s); (**e**) hysteresis loop of the $A_1{}^*$–$D_1{}^*$ relationship (90–150 s); (**f**) hysteresis loop of the $A_1{}^*$–$D_1{}^*$ relationship (at the time of the maximum momentary energy input).

Figure 29 shows the time history of the first modal response for RCDC1 subjected to Uto-NS (Case MF). It is seen that the peak response occurs during UTO0416NS (the first

earthquake). The momentary energy input is a maximum in the half cycle from $t = 28.62$ s to $t + \Delta t = 29.32$ s.

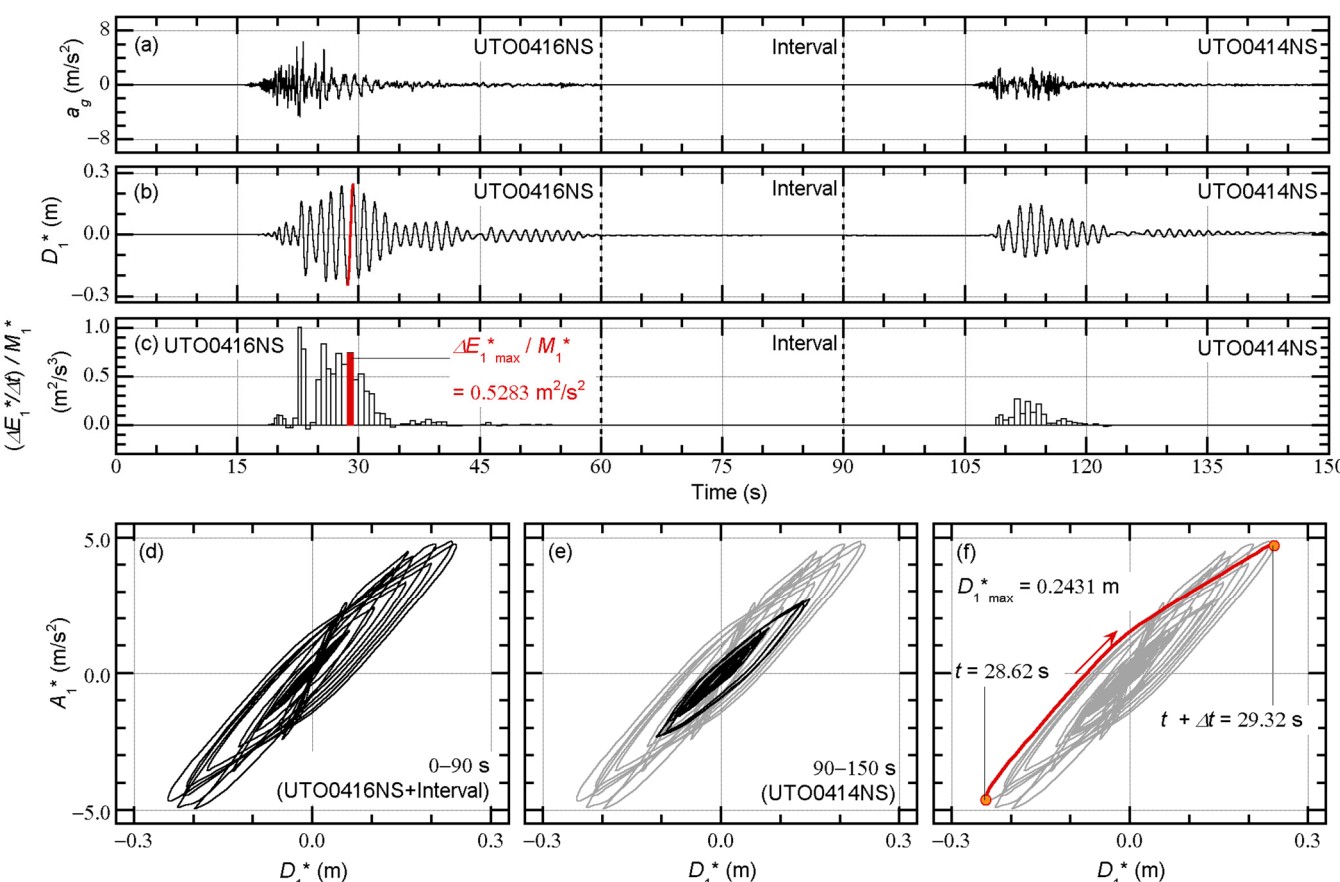

**Figure 29.** Calculated first modal response of RCDC1 subjected to UTO-NS (Case MF): (**a**) time history of the ground acceleration; (**b**) time history of the equivalent displacement; (**c**) time history of the momentary energy input per unit mass; (**d**) hysteresis loop of the $A_1{}^*$–$D_1{}^*$ relationship (0–90 s); (**e**) hysteresis loop of the $A_1{}^*$–$D_1{}^*$ relationship (90–150 s); (**f**) hysteresis loop of the $A_1{}^*$–$D_1{}^*$ relationship (at the time of the maximum momentary energy input).

The following observations are made from the comparisons of Case FM and Case MF.

- The values of the maximum momentary input energy per unit mass ($\Delta E_1{}^*_{max}/M_1{}^*$) are similar in the two cases. $\Delta E_1{}^*_{max}/M_1{}^* = 0.5430$ m$^2$/s$^2$ in Case FM, whereas $\Delta E_1{}^*_{max}/M_1{}^* = 0.5283$ m$^2$/s$^2$ in Case MF.
- The values of the peak equivalent displacement ($D_1{}^*_{max}$) are similar in the two cases. $D_1{}^*_{max} = 0.2597$ m in Case FM, whereas $D_1{}^*_{max} = 0.2431$ m in Case MF.
- The residual equivalent displacement after the first earthquake observed during the interval from 60 to 90 s is small in both cases.
- The hysteresis loop during UTO0414NS acceleration is different. In Case FM (where UTO0414NS acceleration is used for the first earthquake), the hysteresis loop during UTO0414NS acceleration is not thick (i.e., there is little hysteretic energy dissipation). In contrast, the hysteresis loop during UTO0414NS acceleration is thick (i.e., there is much hysteretic energy dissipation) in Case MF (where UTO0414NS acceleration is used for the second earthquake).

Figures 28 and 29 indicate that the order of ground accelerations in sequential accelerations may affect the cumulative response notably, whereas the effect on the peak response may be limited.

Figure 30 compares the peak equivalent displacement ($D_1{}^*{}_{max}$) (a) between Cases F and M, (b) between Cases FM and MF, and (c) between the maximum of Case F, Case M and the maximum of Case FM, Case MF.

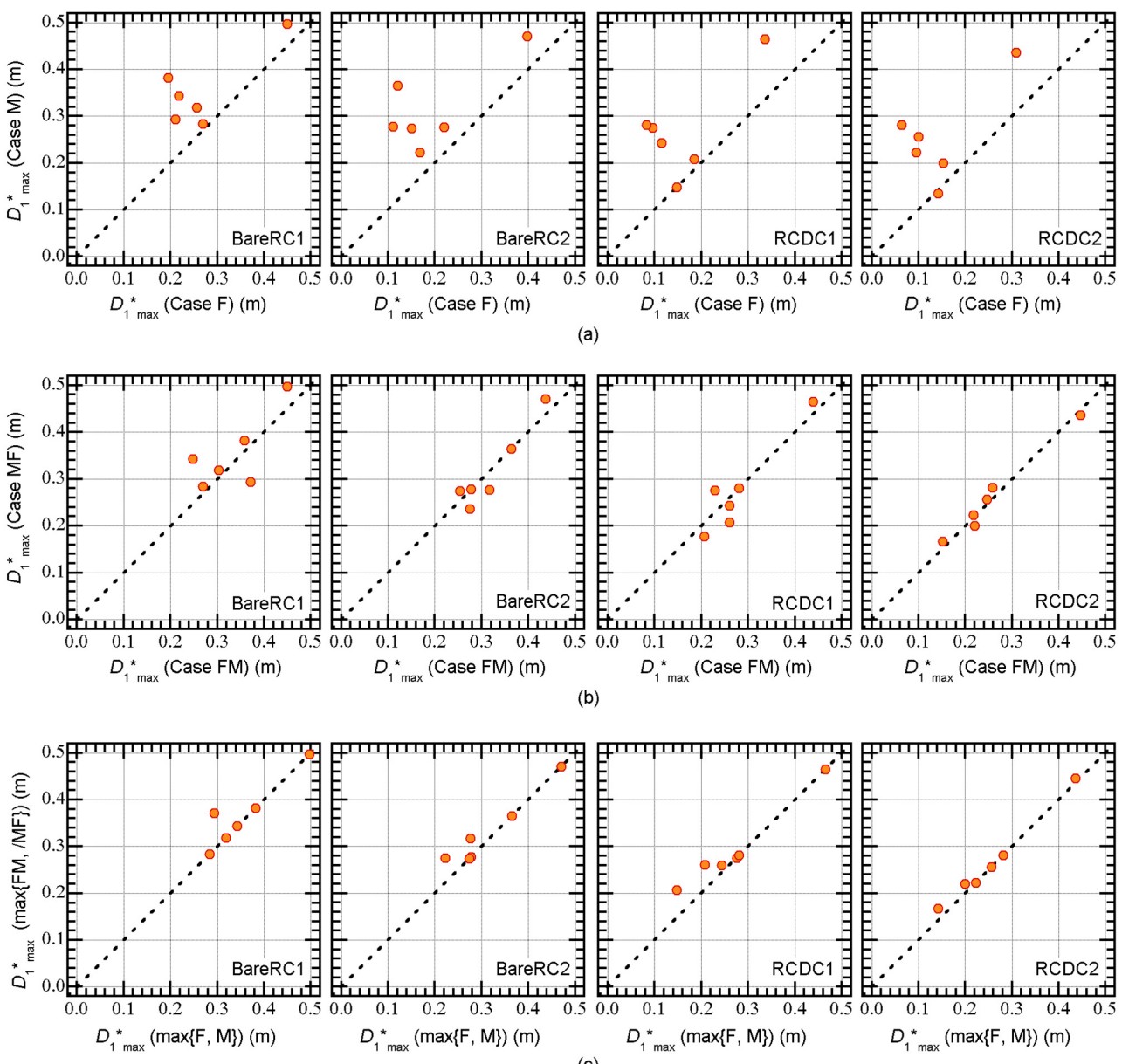

**Figure 30.** Comparisons of the peak equivalent displacement: (**a**) single acceleration; (**b**) sequential acceleration; (**c**) maximum of sequential accelerations and maximum of single accelerations.

The following observations are made for Figure 30.

- According to the single acceleration shown in Figure 30a, the value of $D_1{}^*{}_{max}$ obtained in Case M (mainshock) is larger than that obtained in Case F (foreshock).
- The effect of the order of ground accelerations in sequential accelerations on $D_1{}^*{}_{max}$ is limited, as shown in Figure 30b. The value of $D_1{}^*{}_{max}$ obtained in Case FM is similar to that obtained in Case MF.
- $D_1{}^*{}_{max}$ obtained as the maximum of single accelerations (Case F, Case M) is similar to that obtained as the maximum of sequential accelerations (Case FM, Case MF), as shown in Figure 30c.

To understand the third observation, the residual displacement after the first earthquake is investigated. In this study, the residual equivalent displacement after the first earthquake is defined as the absolute value of the equivalent displacement at $t = t_{end1}$. Here, $t_{end1}$ is defined as the end time of the interval (=90 s). The ratio of the residual equivalent displacement is then defined as

$$r_{resD} = |D_1{}^*(t_{end1})| / D_1{}^*{}_{max}[0, t_{end1}]. \tag{20}$$

In Equation (20), $D_1{}^*{}_{max}[0, t_{end1}]$ is defined as the local peak equivalent displacement within the range $[0, t_{end1}]$. Note that $D_1{}^*{}_{max}[0, t_{end1}]$ equals $D_1{}^*{}_{max}$ obtained from the first earthquake. In Case FM, $D_1{}^*{}_{max}[0, t_{end1}]$ equals $D_1{}^*{}_{max}$ obtained in Case F.

Figure 31 shows the relation between the ratio $r_{resD}$ and the peak equivalent displacement during the first earthquake ($D_1{}^*{}_{max}[0, t_{end1}]$). It is seen that the ratio $r_{resD}$ increases with $D_1{}^*{}_{max}[0, t_{end1}]$. However, in Case FM, the ratio $r_{resD}$ is less than 0.03, except for MAS-EW. Therefore, the effect of the residual displacement of the first earthquake on $D_1{}^*{}_{max}$ is small in Case FM. In addition, the peak equivalent displacement occurs during the first earthquake in Case MF. There is thus no effect of the residual displacement in Case MF, although the ratio $r_{resD}$ is greater than that in Case FM.

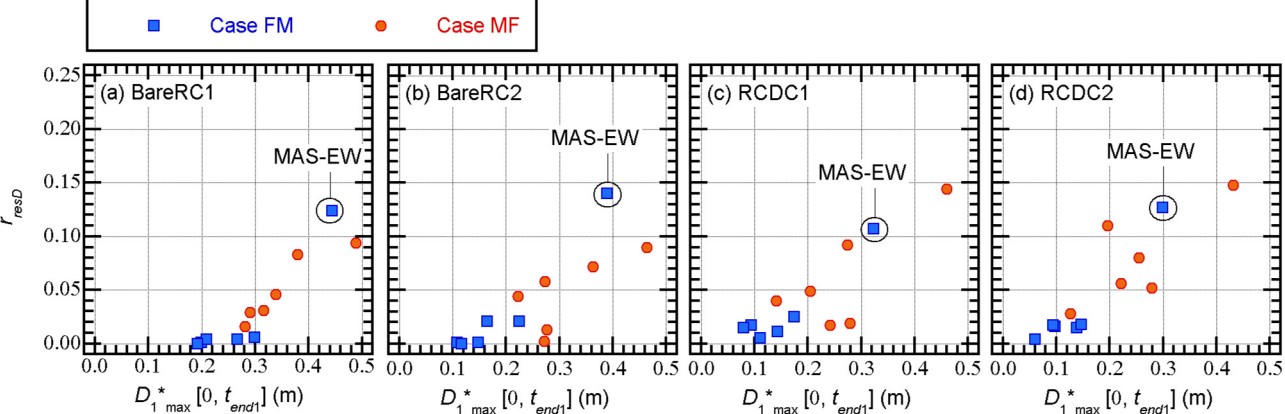

**Figure 31.** Relations of the ratio $r_{resD}$ and $D_1{}^*{}_{max}[0, t_{end1}]$: (**a**) BareRC1; (**b**) BareRC2; (**c**) RCDC1; (**d**) RCDC2.

One reason why the ratio $r_{resD}$ is small (e.g., mostly less than 0.03 in Case FM) is that the unloading slope is degrading as shown in Figures 28 and 29 and was pointed out by Ruiz-García [15].

The discussion moves next to the relation between the maximum momentary energy input and the peak displacement of the first modal response. Figure 32 shows the relation of the equivalent velocity of the maximum momentary input energy ($V_{\Delta E1}{}^*$) and the peak equivalent displacement ($D_1{}^*{}_{max}$). A clear relationship is observed between $V_{\Delta E1}{}^*$ and $D_1{}^*{}_{max}$ for both sequential accelerations and single accelerations; i.e., the peak displacement $D_1{}^*{}_{max}$ increases with $V_{\Delta E1}{}^*$. In addition, the plots obtained for sequential accelerations and single accelerations may be expressed by the same curve. Therefore, the concept of the maximum momentary energy input may be applicable to the prediction of the peak response under sequential accelerations.

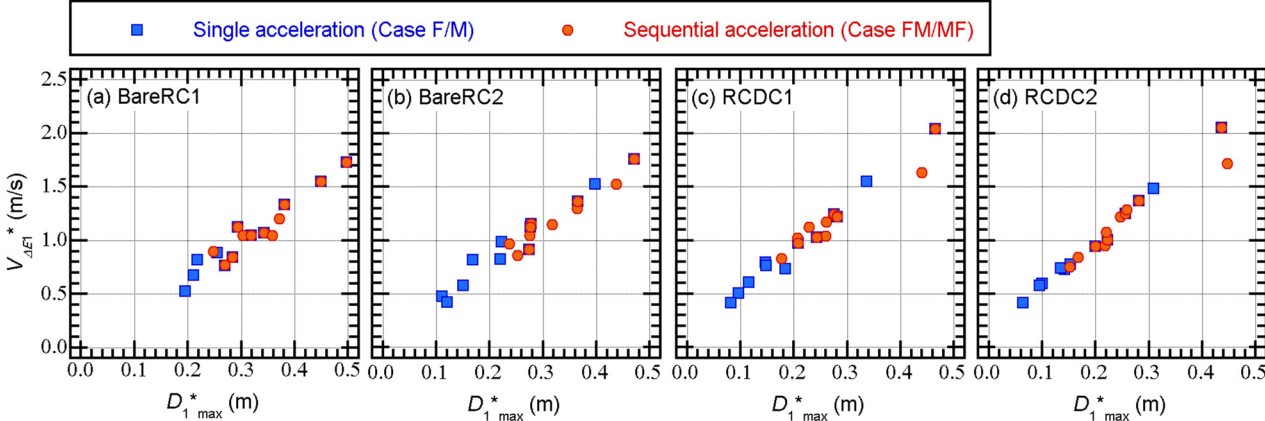

**Figure 32.** Relations of $V_{\Delta E1}{}^*$ and $D_1{}^*{}_{max}$: (**a**) BareRC1; (**b**) BareRC2; (**c**) RCDC1; (**d**) RCDC2.

For the prediction of the peak response of the structure with viscous damping, the effect of viscous damping on the hysteretic energy in a half cycle must be evaluated. To this end, Figure 33 shows the relation between the ratio $V_{\Delta EH1}{}^*/V_{\Delta E1}{}^*$ and the peak equivalent displacement $D_1{}^*{}_{max}$, where $V_{\Delta EH1}{}^*$ is the equivalent velocity of the hysteretic dissipated energy in a half cycle at the maximum momentary energy input (Equations (18) and (19)). The figure shows that the range of the ratio $V_{\Delta EH1}{}^*/V_{\Delta E1}{}^*$ is between 0.80 and 0.97 for BareRC1 and BareRC2 and between 0.90 and 0.98 for RCDC1 and RCDC2. Therefore, the effect of the viscous damping on the ratio $V_{\Delta EH1}{}^*/V_{\Delta E1}{}^*$ is small for the building model with damper columns. This is because the contribution of the stiffness of steel damper columns is excluded when calculating the damping matrix in the nonlinear time-history analysis.

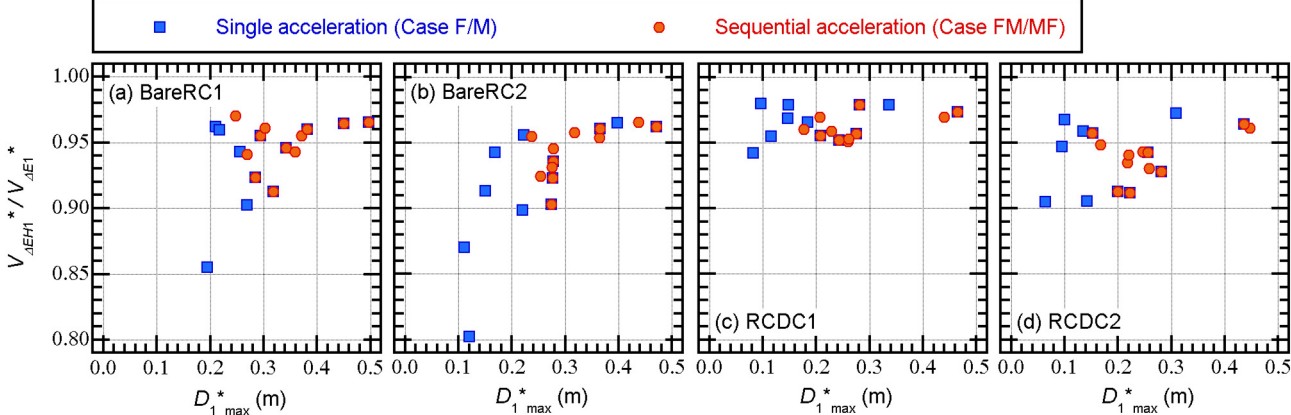

**Figure 33.** Relations of the ratio $V_{\Delta EH1}{}^*/V_{\Delta E1}{}^*$ and $D_1{}^*{}_{max}$: (**a**) BareRC1; (**b**) BareRC2; (**c**) RCDC1; (**d**) RCDC2.

### 4.3. Modeling of the Hysteretic Energy Dissipation of the First Modal Response in a Half Cycle

This section models the hysteresis energy absorption of the first modal response in a half cycle using the results of pushover analysis. The relationship between the equivalent acceleration $A_1{}^*$ and the equivalent displacement $D_1{}^*$ obtained from the results of pushover analysis is first validated by comparing with the first modal responses calculated from the results of nonlinear time-history analysis.

Figure 34 compares the $_nA_1{}^*-_nD_1{}^*$ relationship obtained from the results of pushover analysis (the same as shown in Figure 7) and the plot of the peak response point $(D_1{}^*{}_{max}, A_1{}^*{}_{max})$ obtained from the results of time-history analysis. In Figure 34a,b, the plot peak response points fit well with the results of pushover analysis for BareRC1 and BareRC2. However, there are points plotted below the pushover analysis results, especially for BareRC2. In addition, Figure 34c,d show that the plot peak response points fit

well with the pushover analysis results for RCDC1 and RCDC2, although there are points plotted above the results of pushover analysis. This is because the strain hardening effect of steel damper columns strengthens the restoring force of the overall building models for RCDC1 and RCDC2. It is concluded from Figure 34 that the $_nA_1{}^*{-}_nD_1{}^*$ relationship can be evaluated from the results of pushover analysis.

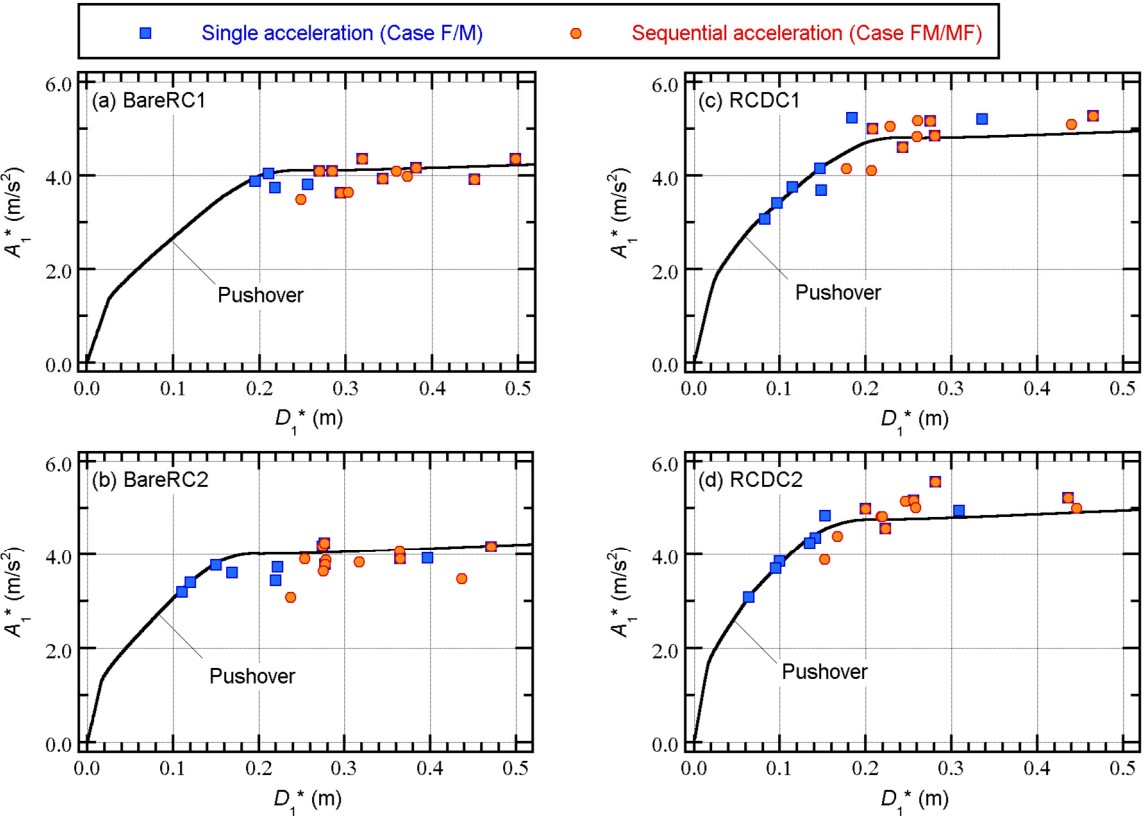

**Figure 34.** Comparisons of the $_nA_1{}^*{-}_nD_1{}^*$ relationship obtained from the results of pushover analysis and the plot of the peak response point $(D_1{}^*{}_{max}, A_1{}^*{}_{max})$ obtained from the results of time-history analysis: (**a**) BareRC1; (**b**) BareRC2; (**c**) RCDC1; (**d**) RCDC2.

Next, the hysteresis energy absorption of the first modal response in a half cycle is modeled using the results of pushover analysis as follows. Let $_n\mathbf{f_{Rf}}$ and $_n\mathbf{f_{Rd}}$ be the restoring force vectors of RC frames and steel damper columns at step $n$ of the pushover analysis, respectively. The contributions of RC frames and steel damper columns to the equivalent acceleration, $_nA_{1f}{}^*$ and $_nA_{1d}{}^*$, respectively, are calculated as

$$_nA_{1f}{}^* = \frac{_n\Gamma_{1n}\boldsymbol{\varphi_1}^T{}_n\mathbf{f_{Rf}}}{_nM_1{}^*} = \frac{_n\mathbf{d}^T{}_n\mathbf{f_{Rf}}}{_n\mathbf{d}^T\mathbf{M1}}, \tag{21}$$

$$_nA_{1d}{}^* = \frac{_n\Gamma_{1n}\boldsymbol{\varphi_1}^T{}_n\mathbf{f_{Rd}}}{_nM_1{}^*} = \frac{_n\mathbf{d}^T{}_n\mathbf{f_{Rd}}}{_n\mathbf{d}^T\mathbf{M1}}. \tag{22}$$

For simplicity, the $_nA_{1f}{}^*{-}_nD_1{}^*$ and $_nA_{1d}{}^*{-}_nD_1{}^*$ relationships are idealized as bilinear curves, where the "yield" point of the idealized $A_{1f}{}^*{-}D_1{}^*$ relationship is $Y_F\left(D_{1yf}{}^*, A_{1yf}{}^*\right)$ and that of the idealized $A_{1d}{}^*{-}D_1{}^*$ relationship is $Y_D\left(D_{1yd}{}^*, A_{1yd}{}^*\right)$. Figure 35 shows the bilinear idealization of the $A_{1f}{}^*{-}D_1{}^*$ relationship for BareRC1 and BareRC2. In the figure, the point $P_1\left(_{P_1}D_1{}^*, _{P_1}A_1{}^*\right)$ is the point at which the first yielding of the RC member occurs, whereas the point $P_2\left(_{P_2}D_1{}^*, _{P_2}A_1{}^*\right)$ is the point at which the equivalent displacement

$_nD_1{}^*$ reaches the assumed displacement limit $D_1{}^*{}_{limit}$ (=0.2833 m). The equivalent acceleration at point $Y_F$ ($A_{1yf}{}^*$) is determined as the equivalent acceleration at point $P_2$ ($_{P_2}A_1{}^*$). The equivalent displacement at point $Y_F$ ($D_{1yf}{}^*$) is then determined as

$$D_{1yf}{}^* = \left(A_{1yf}{}^* / {}_{P_1}A_{1f}{}^*\right) {}_{P_1}D_1{}^* = \left({}_{P_2}A_{1f}{}^* / {}_{P_1}A_{1f}{}^*\right) {}_{P_1}D_1{}^*. \tag{23}$$

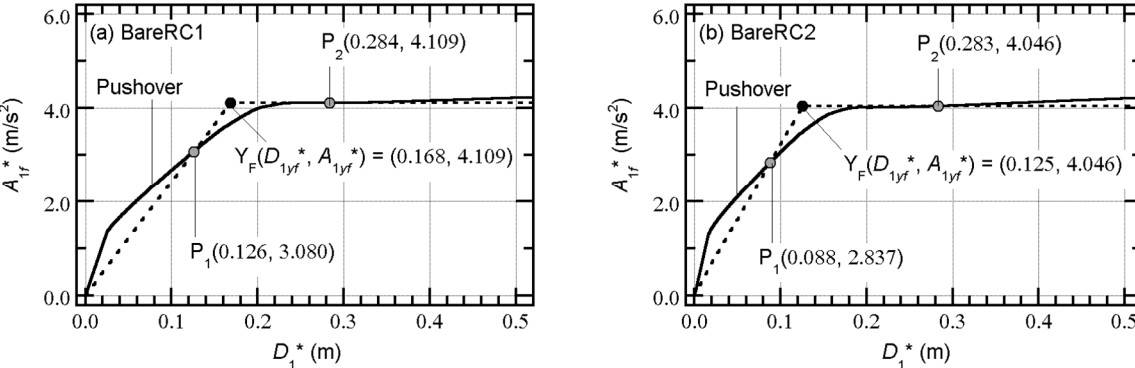

**Figure 35.** Bilinear idealization of the $A_{1f}{}^*{-}D_1{}^*$ relationship from the results of pushover analysis: (**a**) BareRC1; (**b**) BareRC2.

Figure 36 shows the bilinear idealization of the $A_{1f}{}^*{-}D_1{}^*$ and $_nA_{1d}{}^*{-}_nD_1{}^*$ relationships for models RCDC1 and RCDC2. The bilinear idealization is conducted in the same manner for the two relationships. Specifically, the point $P_1$ is taken as the point at which the first yielding of the steel damper column occurs for the bilinear idealization of the $A_{1d}{}^*{-}D_1{}^*$ relationship.

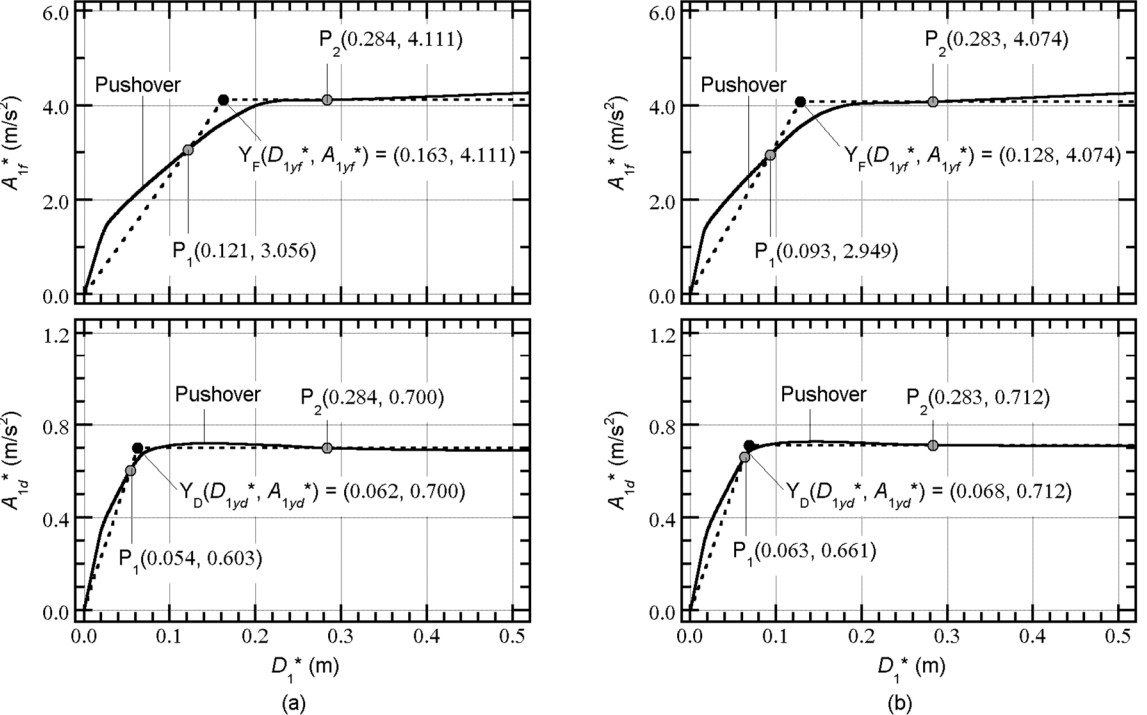

**Figure 36.** Bilinear idealization of the $A_{1f}{}^*{-}D_1{}^*$ and $A_{1d}{}^*{-}D_1{}^*$ relationships from the results of pushover analysis: (**a**) RCDC1; (**b**) RCDC2.

The hysteresis energy absorption of the first modal response in a half cycle per unit mass ($\Delta E_{\mu 1}{}^*/M_1{}^*$) is then formulated as

$$\frac{\Delta E_{\mu 1}{}^*}{M_1{}^*}(D_1{}^*,\eta) = A_{1yf}{}^* D_{1yf}{}^* f_F\left(\mu_f,\eta\right) + A_{1yd}{}^* D_{1yd}{}^* f_D(\mu_d,\eta), \tag{24}$$

$$\mu_f = D_1{}^*/D_{1yf}{}^*, \mu_d = D_1{}^*/D_{1yd}{}^*. \tag{25}$$

In Equation (24), $\eta$ is the ratio of displacements in the positive and negative directions. Figure 37 shows simplified models for calculation of the hysteretic dissipated energy during a half cycle of the structural response. From Figure 37, functions $f_F(\mu_f,\eta)$ and $f_D(\mu_d,\eta)$ are, respectively, calculated as

$$f_F\left(\mu_f,\eta\right) = \begin{cases} \frac{1}{2}\mu_f{}^2\left(1-\eta^2\right) & : \mu_f < 1 \\ \mu_f - \frac{1}{2}\left\{1 + \left(\eta\mu_f\right)^2\right\} & : \mu_f \geq 1 \text{ and } 0 \leq \eta < 1/\mu_f \\ \mu_f - \sqrt{\eta\mu_f} & : \mu_f \geq 1 \text{ and } 1/\mu_f \leq \eta \leq 1 \end{cases}, \tag{26}$$

$$f_D(\mu_d,\eta) = \begin{cases} \frac{1}{2}\mu_d{}^2\left(1-\eta^2\right) & : \mu_d < 1 \\ \mu_d - \frac{1}{2}\left\{1 + (\eta\mu_d)^2\right\} & : \mu_d \geq 1 \text{ and } 0 \leq \eta < 1/\mu_d \\ (1+\eta)\mu_d - 2 & : \mu_d \geq 1 \text{ and } 1/\mu_d \leq \eta \leq 1 \end{cases}. \tag{27}$$

The equivalent velocity of $\Delta E_{\mu 1}{}^*/M_1{}^*$ is defined as

$$V_{\Delta E\mu 1}{}^* = \sqrt{2\Delta E_{\mu 1}{}^*/M_1{}^*}. \tag{28}$$

Figure 38 compares the calculated $V_{\Delta E\mu 1}{}^* - D_1{}^*$ relationship obtained from the results of pushover analysis and the plot of the point $(D_1{}^*{}_{max}, V_{\Delta EH1}{}^*)$ obtained from the results of time-history analysis. In the figure, the ratio of displacements in the positive and negative directions is set as $\eta = 0$, 0.5, and 1.0. It is seen that most of the plots of the point $(D_1{}^*{}_{max}, V_{\Delta EH1}{}^*)$ are within the area bounded by the two $V_{\Delta E\mu 1}{}^* - D_1{}^*$ curves assuming $\eta = 0$ and 1.0, for all four models. Note that for RCDC1 and RCDC2, there are a few points plotted above the $V_{\Delta E\mu 1}{}^* - D_1{}^*$ curve assuming $\eta = 0$ (upper bound curve). This is because the simplified models shown in Figure 37 neglect the strain-hardening effect of the steel damper columns. The comparisons in Figure 38 confirm that the $V_{\Delta E\mu 1}{}^* - D_1{}^*$ relationship of the RC MRF building models with damper columns can be properly evaluated using the simplified model presented herein, for the case of sequential accelerations.

*4.4. Summary of Discussions*

The discussions presented in this section are summarized as follows.

- A method of calculating the first modal response from the results of time-history analysis is proposed. When this method is adopted, the first mode vector is assumed from the time history of the relative horizontal displacement vector.
- The effect of the order of sequential ground accelerations on the peak equivalent displacement of the first modal response ($D_1{}^*{}_{max}$) is limited.
- A clear relationship is observed between the equivalent velocity of the maximum momentary input energy of the first modal response ($V_{\Delta E1}{}^*$) and $D_1{}^*{}_{max}$ for both sequential accelerations and single accelerations; i.e., the peak displacement $D_1{}^*{}_{max}$ increases with $V_{\Delta E1}{}^*$. Therefore, the concept of the maximum momentary energy input may be applicable to the prediction of the peak response under sequential accelerations.
- A simplified model capable of evaluating the hysteretic dissipated energy of the first modal response in a half cycle for given equivalent displacement ($D_1{}^*$) is proposed. The simplified model can be used to evaluate the relationship between the equivalent velocity of the hysteretic dissipated energy in a half cycle ($V_{\Delta E\mu 1}{}^*$) and $D_1{}^*$ with

accuracy. The simplified model is applicable to RC MRFs with and without steel damper columns subjected to sequential accelerations.

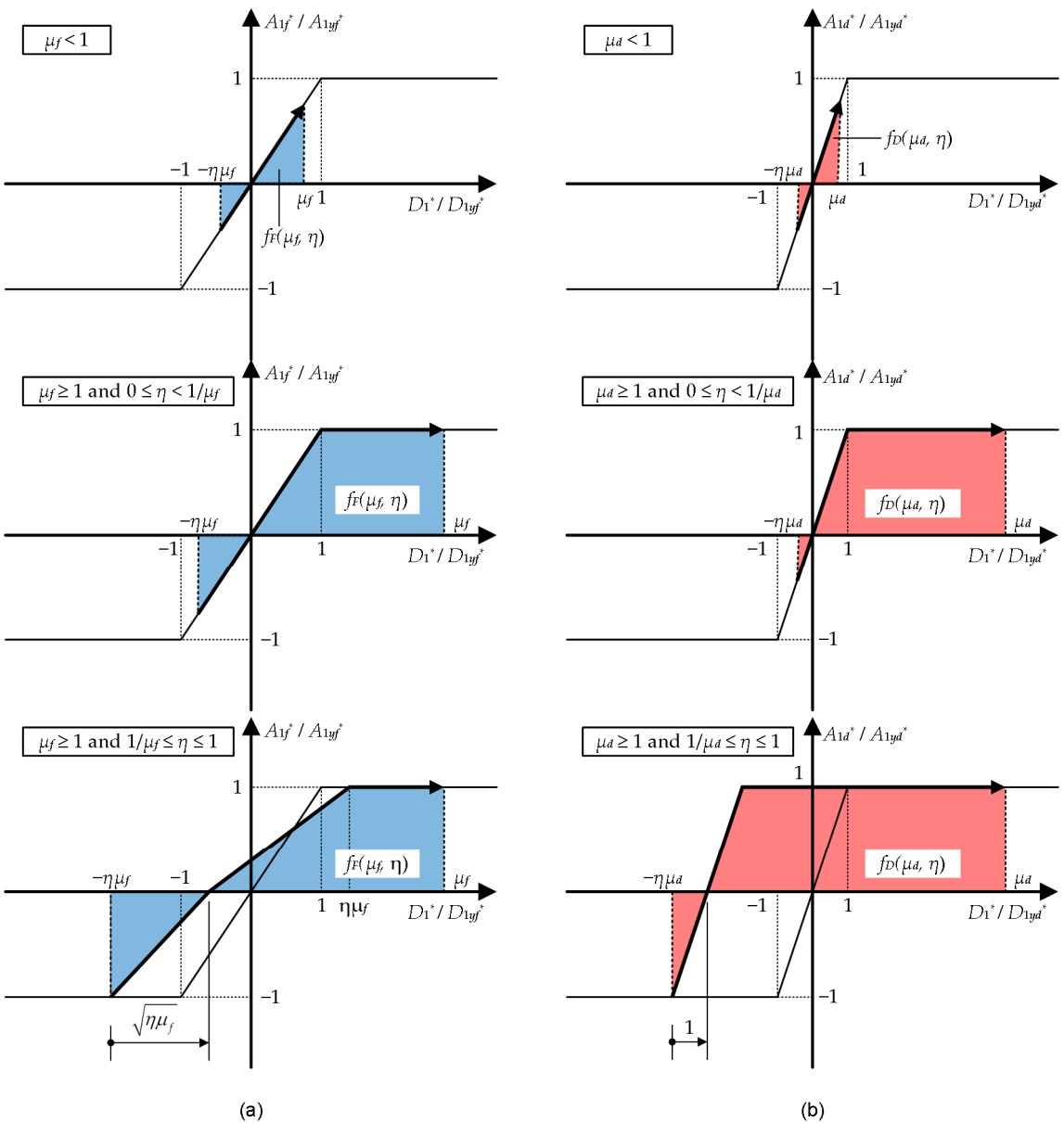

**Figure 37.** Simplified models for calculation of the hysteretic dissipated energy during a half cycle of the structural response: (**a**) contribution of RC MRFs; (**b**) contribution of steel damper columns.

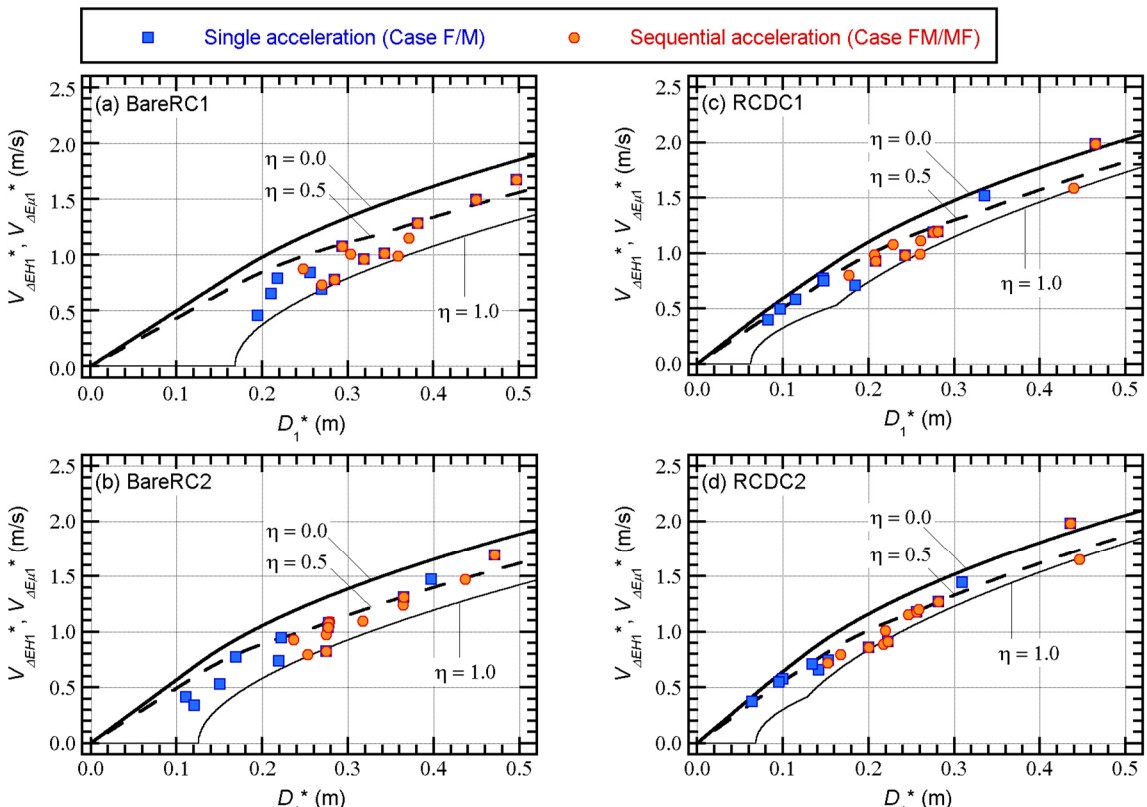

**Figure 38.** Comparisons of the $V_{\Delta E\mu1}{}^*-D_1{}^*$ relationship obtained from the results of pushover analysis and the plot of the point $(D_1{}^*{}_{max}, V_{\Delta EH1}{}^*)$ obtained from the results of time-history analysis: (**a**) BareRC1; (**b**) BareRC2; (**c**) RCDC1; (**d**) RCDC2.

## 5. Conclusions

This article investigated the nonlinear response of 10-story RC MRF building models with steel damper columns designed according to a simplified method [3] as a case study of RC MRFs with steel damper columns subjected to seismic sequences. The main results and conclusions of the study as summarized as follows.

- The peak response of RC MRFs with damper columns subjected to sequential accelerations recorded in the 2016 Kumamoto earthquake is similar to the peak response obtained considering only the mainshock. However, the cumulative strain energy of RC MRFs accumulates in the event of sequential accelerations.
- The steel damper column is effective for reducing the peak and cumulative responses of RC MRFs in the case of sequential seismic input. The results of nonlinear time-history analysis presented in this study indicate that the installation of steel damper columns can reduce the story drift of RC MRFs and the peak plastic rotation and cumulative strain energy of RC beam ends. However, the designer and structural engineer should pay attention to the behavior of short-span beams in the presence of steel damper columns; i.e., the use of a short-span beam may result in severe damage if its energy absorption capacity is insufficient.
- The relation of the hysteretic dissipated energy during a half cycle of the structural response and the peak displacement of the first modal response can be properly evaluated using the simple model proposed in this study. The proposed simple model can be applied for RC MRFs with and without hysteresis dampers.

It is obvious that the conclusions drawn in this article are valid only for the limited conditions of four 10-story RC MRFs and six sets of seismic sequences recorded during the foreshock and mainshock of the 2016 Kumamoto earthquake. Despite such limitations, three points can be made:

- The first point is that the results presented in Section 3 emphasize the importance of the cumulative response of the structures in the case of seismic sequences. Unlike the peak deformation of the members, the cumulative strain energy accumulates in the event of seismic sequences. The evaluation of the cumulative response is important because the peak deformation and the cumulative strain energy are essential parameters for evaluating the structural damage to members.
- The second point is that the method of calculating the first model response presented here is effective for the post-analysis of nonlinear time-history analysis. This calculation can help the analyst further understand the nonlinear response; e.g., the calculated first modal response can be compared directly with the response of the idealized SDOF model. Note that this calculation is applicable to the post-analysis of the nonlinear time-history analysis and also experimental results, provided the building considered oscillates in the first mode.
- The third point is that the simplified modeling of the hysteretic dissipated energy during a half cycle of the structural response for a given peak equivalent displacement discussed in Section 4 is essential to the prediction of the peak response of RC MRFs with steel damper columns. Using the maximum momentary input energy spectrum introduced by Inoue and his coauthors [59–61], the peak equivalent displacement of the first modal response can be predicted.

Note that the cyclic degradation of members in RC MRFs studied herein is relatively mild. The strength degradation is not considered because there is shear reinforcement in all RC members sufficient to prevent shear failure. However, if both the stiffness and strength degradation due to cyclic loading are notable, the simplified model proposed in Section 4 may need to be revised to include the effect of cyclic degradation. In addition, the effect of cyclic loading would be more pronounced in the case of seismic sequences with long durations, e.g., the records of the 2011 Tohoku earthquake in Japan. In the RC MRF with steel damper columns studied herein, the effectiveness of steel damper columns may deteriorate if the short-span RC beams connected to damper columns are severely damaged. Such issues will be investigated in the next phase of this study.

**Funding:** This research received financial support from JFE Civil and Construction Corp.

**Institutional Review Board Statement:** Not applicable.

**Informed Consent Statement:** Not applicable.

**Data Availability Statement:** The data presented in this study are available on request from the corresponding author.

**Acknowledgments:** The original frame model data used in this study were provided by Riho Mukouyama, a former graduate student of Chiba Institute of Technology. Valuable comments from Chizuru Irie and Mitsuhide Yoshinaga, JFE Civil Engineering, and Construction Corp. are appreciated. The ground motions used in this study were obtained from the website of the National Research Institute for Earth Science and Disaster Resilience (NIED) (https://www.kyoshin.bosai.go.jp/, last accessed on 7 January 2022).

**Conflicts of Interest:** The funders had no role in the design of the study; in the collection, analyses, or interpretation of data; in the writing of the manuscript; or in the decision to publish the results.

**Appendix A. Definition of the Momentary Input Energy**

Here, the definition of the momentary input energy is described. Consider the SDOF model with mass ($m$), damping force ($f_D$), and restoring force ($f_R$). The equation of motion of the SDOF model can be written as

$$m\ddot{y} + f_D + f_R = -ma_g. \tag{A1}$$

By multiplying both sides of Equation (A1) by $\dot{y}dt$ and integrating from 0 to $t$, the equation of energy balance from time 0 to $t$ is obtained as

$$E_V(t) + E_D(t) + E_H(t) = E_I(t), \tag{A2}$$

$$E_V(t) = \int_0^t m\ddot{y}\dot{y}dt, \, E_D(t) = \int_0^t f_D\dot{y}dt, \, E_H(t) = \int_0^t f_R\dot{y}dt, \, E_I(t) = -\int_0^t ma_g\dot{y}dt. \tag{A3}$$

Here, $E_V(t)$ is the kinetic energy, $E_D(t)$ is the damping dissipated energy, $E_H(t)$ is the hysteretic dissipated energy, and $E_I(t)$ is the (relative) input energy.

Next, the momentary input energy is defined as follows. Following the work of Inoue and his coauthors [59–61], we consider the energy balance during a half cycle of structural response (from $t$ to $t + \Delta t$). The beginning and end times of a half cycle, $t$ and $t + \Delta t$, respectively, are defined as times when the displacement ($y(t)$) is a local maximum. The equation of the energy balance during a half cycle of structural response is expressed as

$$\Delta E_V + \Delta E_D + \Delta E_H = \Delta E, \tag{A4}$$

$$\begin{cases} \Delta E_V = \int\limits_t^{t+\Delta t} m\ddot{y}\dot{y}dt = \left[\frac{1}{2}m\dot{y}^2(t)\right]_t^{t+\Delta t}, \Delta E_D = \int\limits_t^{t+\Delta t} f_D\dot{y}dt, \\ \Delta E_H = \int\limits_t^{t+\Delta t} f_R\dot{y}dt, \Delta E = -\int\limits_t^{t+\Delta t} ma_g\dot{y}dt \end{cases} \tag{A5}$$

In Equation (A4), $\Delta E_V = 0$ because the velocities at time $t$ and $t + \Delta t$ are zero ($\dot{y}(t) = \dot{y}(t + \Delta t) = 0$). Therefore, Equation (A4) can be rewritten as

$$\Delta E_D + \Delta E_H = \Delta E. \tag{A6}$$

Equation (A6) indicates that the input energy during a half cycle ($\Delta E$) equals to the sum of the damping dissipated energy ($\Delta E_D$) and hysteretic dissipated energy ($\Delta E_H$). Based on discussions above, we denote the input energy during a half cycle ($\Delta E$) as the momentary input energy and consider as the seismic intensity parameter related to the peak response.

**Appendix B. Model Properties**

Here, the properties of the members of the four models analyzed in this study are described. Note that the properties of the models RCDC1 and BareRC1 are taken from previous studies [3,4] with some updates.

Table A1 presents the sections at potential hinges of RC members of RCDC1 and BareRC1. Note that the cross-sections of all RC columns have dimensions of 900 mm $\times$ 900 mm, which are the same as those of the cross-section at the bottom of the first story. The cross section of the RC beams at the foundation level ($Z_0$) has dimensions of 800 mm $\times$ 1900 mm. The yield strength of the longitudinal reinforcement is assumed to be $1.1 \times 390 = 429$ N/mm². The assumed compressive strength of concrete is 33 N/mm² for the first and second stories, 30 N/mm² for the third to fifth stories, and 27 N/mm² at and above the sixth story.

Table A2 presents the selected damper columns of RCDC1. The initial normal yield stress of the steel used for the damper panel is assumed to be 205 N/mm², whereas the normal yield stress after appreciable cyclic loading is assumed to be 300 N/mm².

Table A3 presents the sections at potential hinges of RC members for RCDC2 and BareRC2. Note that the cross-sections of all RC columns have dimensions of 800 mm $\times$ 800 mm, which are the same as those of the cross section at the bottom of the first story. The cross section of the RC beams at the foundation level ($Z_0$) has dimensions of 800 mm $\times$ 1900 mm. The yield strength of the longitudinal reinforcement and the assumed compressive strength of concrete are the same as for RCDC1 (BareRC1).

Table A4 presents the selected damper columns of RCDC2. The properties of the steel used for the damper panel are the same as those for RCDC1.

**Table A1.** Sections at potential hinges of RC members for RCDC1 and BareRC1.

| Member | Location | Width (mm) | Depth (mm) | Longitudinal Reinforcement |
|---|---|---|---|---|
| Beam | $Z_6$ to $Z_{10}$ | 500 | 900 | 10-D29 (Top and bottom) |
| | $Z_5$ | 550 | 900 | 8-D32 (Top and bottom) |
| | $Z_4$ | 600 | 900 | 9-D32 (Top and bottom) |
| | $Z_2$ to $Z_3$ | 600 | 900 | 8-D35 (Top and bottom) |
| | $Z_1$ | 800 | 900 | 9-D38 (Top and bottom) |
| Column | 1st Story (Bottom) | 900 | 900 | 10-D38 |

**Table A2.** Steel damper columns for RCDC1.

| Story | Yield Strength | | Panel Thickness (mm) | Panel Sectional Area (mm$^2$) | Column (mm $\times$ mm $\times$ mm $\times$ mm) |
|---|---|---|---|---|---|
| | $Q_{yDL}$ (kN) | $Q_{yDU}$ (kN) | | | |
| 10 | 438 | 641 | 6 | 3700 | H-600 $\times$ 200 $\times$ 12 $\times$ 25 |
| 9 | 626 | 916 | 9 | 5290 | H-600 $\times$ 250 $\times$ 16 $\times$ 32 |
| 8 | 755 | 1105 | 9 | 6380 | H-700 $\times$ 300 $\times$ 16 $\times$ 28 |
| 7 | 968 | 1417 | 9 | 8180 | H-900 $\times$ 300 $\times$ 16 $\times$ 28 |
| 5 to 6 | 1251 | 1831 | 9 | 10,580 | H-600 $\times$ 250 $\times$ 16 $\times$ 32 (Doubled) |
| 1 to 4 | 1551 | 2211 | 9 | 12,760 | H-700 $\times$ 300 $\times$ 16 $\times$ 32 (Doubled) |

**Table A3.** Sections at potential hinges of RC members for RCDC2 and BareRC2.

| Member | Location | Width (mm) | Depth (mm) | Longitudinal Reinforcement |
|---|---|---|---|---|
| Beam | $Z_6$ to $Z_{10}$ | 500 | 900 | 7-D29 (Top and bottom) |
| | $Z_5$ | 550 | 900 | 7-D29 (Top and bottom) |
| | $Z_4$ | 600 | 900 | 6-D32 (Top and bottom) |
| | $Z_2$ to $Z_3$ | 600 | 900 | 10-D25 (Top and bottom) |
| | $Z_1$ | 800 | 900 | 9-D32 (Top and bottom) |
| Column | 1st Story (Bottom) | 800 | 800 | 8-D35 |

**Table A4.** Steel damper columns for RCDC2.

| Story | Yield Strength | | Panel Thickness (mm) | Panel Sectional Area (mm$^2$) | Column (mm $\times$ mm $\times$ mm $\times$ mm) |
|---|---|---|---|---|---|
| | $Q_{yDL}$ (kN) | $Q_{yDU}$ (kN) | | | |
| 9 to 10 | 438 | 641 | 6 | 3700 | H-600 $\times$ 200 $\times$ 12 $\times$ 25 |
| 7 to 8 | 626 | 916 | 9 | 5290 | H-600 $\times$ 250 $\times$ 16 $\times$ 32 |
| 6 | 755 | 1105 | 9 | 6380 | H-700 $\times$ 300 $\times$ 16 $\times$ 28 |
| 5 | 862 | 1261 | 9 | 7280 | H-800 $\times$ 300 $\times$ 16 $\times$ 28 |
| 2 to 4 | 968 | 1417 | 9 | 8180 | H-900 $\times$ 300 $\times$ 16 $\times$ 28 |
| 1 | 1251 | 1841 | 12 | 10,630 | H-900 $\times$ 300 $\times$ 19 $\times$ 32 |

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
