# Peer review of "Peak and Cumulative Response of Reinforced Concrete Frames with Steel Damper Columns under Seismic Sequences"

_buildings, doi:10.3390/buildings12030275_

Round 1
Reviewer 1 Report
It's interesting that this manuscript focuses on peak and cumulative response of reinforced concrete frames with steel damper columns under seismic sequences.
However, the dynamic input energy, such as momentary input engrgy, should be introduced in detail in this manuscript, which is the driving force mainly acting on the framework.
The reinforcement of beams and columns in the frame should be introduced in detail in this manuscript.
The connection design of beam column joints in frames should be introduced in detail in this manuscript.
Some of the research addressing these issues should be acknowledged, some recommended references, among many others are, https://doi.org/10.1016/j.istruc.2020.12.089;https://doi.org/10.1061/(ASCE)AS.1943-5525.0000942; https://doi.org/10.1002/tal.1775; https://doi.org/10.1016/j.jseaes.2013.05.008; https://doi.org/10.3311/PPci.15276; and so on.
Once the above concerns are fully addressed, I would be very glad to re-review the manuscript in greater depth because the subject is interesting.
Reviewer 2 Report
Impressive work. The manuscript is much longer than average, perhaps the author may consider splitting it into two papers.
Reviewer 3 Report
Comments and Suggestions for Authors in the attached pdf file.

Round 2
Reviewer 1 Report
The authors have NOT addressed most of the comments raised by the reviewers substantially.
Due to the those fatal defects, it is difficult for readers to verify the correctness, accuracy and reliability of this manuscript.
Therefore, it is considered that this manuscript is not suitable for publication in this international journal.
Reviewer 3 Report
The author did not fully take into account all the comments of this reviewer especially with reference to point 5 of the review report. Thus, that paper cannot be accepted in such a version and it is necessary another round of review.
Round 3
Reviewer 3 Report
The authors made the required corrections. The paper can be accepted.